# ComputAgeBench: Epigenetic Aging Clocks Benchmark

## Abstract

The success of clinical trials of longevity drugs relies heavily on identifying integrative health and aging biomarkers, such as biological age. Epigenetic aging clocks predict the biological age of an individual using their DNA methylation profiles, commonly retrieved from blood samples. However, there is no standardized methodology to validate and compare epigenetic clock models as yet. We propose ComputAgeBench, a unifying framework that comprises such a methodology and a dataset for comprehensive benchmarking of different clinically relevant aging clocks. Our methodology exploits the core idea that reliable aging clocks must be able to distinguish between healthy individuals and those with aging-accelerating conditions. Specifically, we collected and harmonized 66 public datasets of blood DNA methylation, covering 19 such conditions across different ages and tested 13 published clock models. We believe our work will bring the fields of aging biology and machine learning closer together for the research on reliable biomarkers of health and aging.

## 1 Introduction

Longevity drugs (*a.k.a., geroprotectors*) appear to be on the brink of entering clinical practice to slow down or reverse the features of aging (Moqri et al., 2024; Justice et al., 2018). The research community is yet to identify proper biomarkers of aging and rejuvenation that could be used as clinical trial endpoints instead of or in combination with observations on patient lifespans (Schork et al., 2022). Biological age (BA) has been proposed as one of such surrogate biomarkers of aging, defined as a *generalized measure of human health* compared to the average health of individuals at a given age within a population (Yousefi et al., 2022; Jylhävä et al., 2017). Thus, if an individual has a biological age of 40 at the chronological age of 30, it is assumed that their overall health corresponds to that of an average 40-year-old in the population. This relationship can be concisely expressed as

$$B = C + \Delta, \tag{1}$$

where $B$ represents biological age, $C$ denotes chronological age (*i.e.*, time since birth), and $\Delta$ symbolizes BA *acceleration* (or deceleration, if negative).

In general, BA can be estimated from a set of biomarkers $X$ with a model (algorithm) $f : X \to B$, also called an *aging clock*. However, BA is latent: it has no ground truth value that can be measured directly and then used to train an aging clock model $f$ in a classical supervised fashion, making clock validation a nontrivial task (Sluiskes et al., 2024). This obstacle forces researchers to introduce various additional assumptions about the aging clock behavior (Klemera & Doubal, 2006; Horvath, 2013; Pierson et al., 2019; Rutledge et al., 2022), as well as to experiment with different machine learning models (including penalized linear regressions, such as ElasticNet, support vector machines, decision trees, transfomer-based neural networks, *etc*. (Rutledge et al., 2022; Urban et al., 2023)) and underlying types of data $X$ (Putin et al., 2016; Xia et al., 2020; Holzscheck et al., 2021). The vast majority of aging clocks, though, rely primarily on DNA methylation data, also called *epigenetic* aging clocks (Hannum et al., 2013; Levine et al., 2018; Lu et al., 2019; Galkin et al., 2021; Ying et al., 2024). Summarizing abundant discussions about a "good" mathematical description of BA in the literature (Moskalev, 2019; Rutledge et al., 2022; Moqri et al., 2024), we elicited four of its defining properties, formalized as follows.

Let $X \in \mathbb{R}^p$, where $p$ is the number of biomarkers in data, $B \in \mathbb{R}$, and $f : X \to B$. Given the aging acceleration $\Delta = B - C$, the following four properties hold:

Figure 1: ComputAgeBench: benchmarking various epigenetic aging clock models. For a dataset $X$, obtained by profiling DNA methylation at CpG sites in bulk blood samples, an aging clock model $f$ is trained to distinguish healthy individuals from those with pre-defined aging-accelerating conditions.

1. $B$ is expressed in the same time units as $C$;
2. $\Delta$ allows distinguishing between healthy individuals and individuals with aging-accelerating or decelerating conditions (AACs or ADCs), such as severe chronic diseases;
3. $B$ helps to predict the remaining lifespan better than $C$ does (Moskalev, 2019);
4. $B$ helps to predict the time to onset of chronic age-related diseases (*e.g.*, the Alzheimer's) better than $C$ does (Moskalev, 2019).

Garnered together, these properties motivated us to construct a benchmarking methodology for validating the potential biological age predictors. In property #1, the model $f$ should output age values in a biologically meaningful range, comparable with a typical lifespan, *e.g.*, from 0 to 120 years for the humans. To investigate if a model $f$ satisfies the $2^{nd}$ property, we can define a panel of aging-accelerating (or decelerating) conditions and test if the predicted $\Delta$ allows distinguishing the individuals with an AAC/ADC from a control group, according to an appropriate statistical test. To validate the compliance with the $3^{rd}$ and the $4^{th}$ properties, one also needs data on mortality and multi-morbidity. That is, the information about the timing of death or the onset of chronic age-related diseases, along with a prior measurement of a set of relevant biomarkers. It is important to note that such data are highly sensitive and are generally not publicly available.

*DNA methylation* (DNAm) is the most prevalent measurement employed in the construction of aging clocks (Xia et al., 2021). From a chemical point of view, DNA methylation refers to a covalent modification of DNA nucleotides by the methyl groups (Greenberg & Bourc'his, 2019). Specifically, cytosine nucleotides (C) followed by guanine nucleotides (G), also referred to as cytosines in a CpG context or simply CpG sites (CpGs), are methylated most often in the mammalian cells, making it the most well-studied type of DNA methylation (Seale et al., 2022) (refer to Fig. 1 for visualization of the DNA and CpGs). This epigenetic modification plays a crucial role in regulating gene expression and is engaged in a variety of cellular events, varying significantly across different species, tissues, and the lifespan. DNA methylation levels per site are usually reported quantitatively as beta values that represent the methylation proportion at a specific CpG site in the range from 0 to 1, where 0 indicates no methylation, and 1 indicates complete methylation across all the cells in the sample (Fig. 1).

Importantly, despite the numerous recent publications of various aging clocks (Xia et al., 2021; Rutledge et al., 2022; Yousefi et al., 2022), including the ones built on DNA methylation, no systematic open access benchmark, which would include standardized panel of datasets, diseases, interventions, or other conditions, has been proposed to date to validate the aforementioned properties. In this paper, we introduce such a benchmark to validate the $1^{st}$ and the $2^{nd}$ properties in epigenetic aging clocks. To do this, we developed a methodology for identifying aging-accelerating conditions, which relies on simple, yet strict and evidence-based principles for defining and selecting a panel of aging-accelerating conditions. We collected an unprecedented number of DNA methylation datasets for the respective conditions from dozens of published studies. We also developed a cumulative benchmarking score that aggregates two error-based tasks and two simple, but informative tasks based on common statistical tests. Ultimately, this cumulative score enables comparing aging clock ability to satisfy the $1^{st}$ and the $2^{nd}$ properties.

To demonstrate our methodology in a clinically relevant scenario, we specifically focused on the blood-, saliva-, and buccal-based epigenetic biomarkers obtained via a microarray-based technology. Such biomarkers are widespread in clinical testing and aging clock construction (Campagna et al., 2021; Rutledge et al., 2022). We then examined 13 published epigenetic clocks and provided their benchmarking results.

## 2 RELATED WORK AND BACKGROUND

### 2.1 AGING CLOCK CONSTRUCTION METHODOLOGY

Because the BA ground truth values cannot be measured, and, therefore, a direct validation of aging clocks is problematic, previous studies introduced various approaches to construct aging clocks with different underlying assumptions. The most widespread one, belonging to the so-called "first-generation aging clocks", uses an assumption that a model $f$ can be trained to predict chronological age, *i.e.*, $C = \hat{C} + \varepsilon = f(X) + \varepsilon$, and its predictions will correspond to BA: $B = \hat{C}$. The simplicity of this approach has made it attractive for decades, and it is still used today to train new aging clocks on new types of data (Hollingsworth et al., 1965; Voitenko & Tokar, 1983; Duggirala et al., 2002; Varshavsky et al., 2023; Prosz et al., 2024). In fact, BA obtained by this approach can satisfy the $2^{nd}$ (Horvath, 2013) and the $3^{rd}$ (Kuiper et al., 2023) properties from our definition. However, using this assumption in Eq. (1) leads us to the conclusion that $\varepsilon = -\Delta$. It then turns out that the perfect solution of the chronological age prediction problem, *i.e.*, minimizing the prediction error so that $\varepsilon \to 0$, leads to the inability of a clock to identify any aging acceleration or deceleration. Namely, it implies that $\Delta \to -0$, which is also known as *the biomarkers paradox* (Hochschild, 1989; Klemera & Doubal, 2006). Supporting this concept, it has been shown that the clocks featuring strong correlation with the chronological age poorly correlate with the population mortality (Zhang et al., 2019) (hence they fail to satisfy the $3^{rd}$ property). As a consequence, validating clock performance in terms of accuracy of chronological age prediction becomes meaningless, because high accuracy may not necessarily correspond to a biologically relevant clock. Despite the obvious methodological challenges of this approach, it is worth noting that the vast majority of aging clocks belong to the first generation (Sluiskes et al., 2024).

Seeking for a better solution, researchers experimented with survival models, which led to the development of "second-generation aging clocks". In this approach, models are trained to predict time to death (Levine et al., 2018; Lu et al., 2019; Hertel et al., 2016), and the resulting prediction is rescaled to age units to represent BA, therefore addressing the $3^{rd}$ and the $4^{th}$ properties of a "good" BA estimator. However, there is no open large-scale DNA methylation data containing time-to-death or multi-morbidity measurements, with existing studies being either available upon an authorized request or being held completely private (see Appendix A.7).

### 2.2 ATTEMPTS TO COMPARE EPIGENETIC AGING CLOCKS

Despite reported attempts to compare the performance of different aging clocks, a benchmark with a standardized panel of datasets, diseases, interventions, or other conditions has not been proposed yet. As a result, different comparative studies employ widely varying validation data and approaches (Moqri et al., 2024; Ying et al., 2024; Kuiper et al., 2023; Mei et al., 2023; Wang et al., 2021; Huan et al., 2022; Chervova et al., 2022; Liu et al., 2020; Maddock et al., 2020; McCrory et al., 2021). As highlighted in a recent review on biomarker validation by Moqri et al. (2024), "*for a reliable comparison across studies, . . . biomarker formulations should be established 'a priori' and not be further modified during validation*". In the same line of thought, we propose to define a standardized and a justified procedure for clock benchmarking *before* constructing any predictive model.

Two approaches we propose as essential tasks in our benchmark entail related prior art. For example, Porter et al. (2021) and Mei et al. (2023) used one-sample or two-sample aging acceleration tests for clock validation. Ying et al. (2024) employed two-sample tests across multiple aging clocks. These authors implicitly tested the $2^{nd}$ property of "good" aging clocks discussed above. Likewise, there were also attempts to test the $3^{rd}$ and the $4^{th}$ properties separately. In other works, including the recently updated pre-print of Biolearn (Ying et al., 2023), a Python-based framework for aging clock training and testing in ongoing development, authors performed Cox Proportional Hazards analysis and calculated hazard ratios with statistical significance to test if BA estimates of selected clocks are

capable of predicting all-cause mortality or the onset of age-related diseases (*e.g.*, cardiovascular events) (Kuiper et al., 2023; Wang et al., 2021; McCrory et al., 2021; Huan et al., 2022; Chervova et al., 2022; Ying et al., 2023). However, these prior studies are either small-scale (Ying et al., 2024), limited to predicting the chronological age (Liu et al., 2020), or miss standardized datasets and compare only a small number of models (Porter et al., 2021; Mei et al., 2023), or rely on mortality and disease data that are under restricted access (Ying et al., 2023). Therefore, while developing our methodology, we attempted to mitigate all mentioned drawbacks.

## 3 BENCHMARKING METHODOLOGY

An infographic overview of the proposed benchmarking of aging clocks is shown in Fig. 2.

### 3.1 CRITERIA FOR SELECTING AGING-ACCELERATING CONDITIONS

In the context of clock benchmarking, we propose to define an aging-accelerating condition (AAC) as a biological condition that satisfies the following three criteria (Fig. 2B). First, having an AAC must lead to decreased life expectancy (LE) compared to the general population, even when treated with existing therapies. Second, an AAC must be chronic (to safely assume that it has sufficient time to drive observable changes in DNAm). And third, an AAC must manifest itself systemically, so that it can be expected to affect DNAm in blood, saliva, and buccal cells (hereafter referred to as BSB).

Importantly, the decrease in LE and the corresponding increase in mortality must result mainly from intrinsic organismal causes rather than from socioeconomic factors and self-destructive behaviors related to a given condition. The second criterion is aimed at excluding short-term conditions such as acute infectious diseases, stressful events, and other confounding DNAm-alternating accidents, whose effects might not induce significant changes in DNAm data obtained from BSB, or, on the contrary, might last too briefly to be reliably detected. The third part of the AAC definition precludes us from considering events with long-lasting and life-threatening consequences that might be difficult to observe in BSB-derived data. For instance, a bone fracture (unless it is a critical bone marrow reserve) or some types of malignancies.

Conversely, an aging-decelerating condition (ADC) is defined as a condition that increases LE, compared to the general population, and features the same second and third criteria as an AAC. With human data, however, the ADCs are difficult to determine, as the human lifespan-increasing interventions are yet to emerge. To avoid ambiguous interpretation, we omitted such conditions in our benchmarking of human aging clocks (see Appendix A.5 and Table A2 for more details).

### 3.2 CRITERIA FOR DATASET SELECTION

Aiming to provide a comprehensive, easily accessible, and clinically relevant toolbox for the ongoing research on human epigenetic clocks, we relied on the following five criteria while performing the datasets aggregation (Fig. 2C). *First*, all datasets in the benchmark must feature *open access to pre-processed data*, without any data access requests or raw data processing required. *Second*, we only used data obtained from the BSB samples. *Third*, chronological ages must be annotated with, at most, one year intervals (*e.g.*, without age binning by decades), including only samples from the age range of 18–90 years[1]. The only exception to this requirement are the individuals with certain *progeroid* conditions, such as the Hutchinson-Gilford progeria syndrome, who survive approximately 12 to 13 years on average: these conditions resemble premature aging so strikingly (Schnabel et al., 2021) that we included patients aged under 18 years into the benchmark. *Fourth*, we employ data obtained only with the Illumina Infinium BeadChip (27K, 450K, and 850K) methylation microarrays, as they remain to be the most popular technologies for human DNAm profiling and clock construction. *Fifth*, we applied thresholds of at least 10 samples per dataset, 5 samples with an AAC per dataset, and 10 samples with an AAC across all datasets to attain sufficient statistical power.

---

[1]Reporting increased or decreased biological age for people outside of this range is debatable.

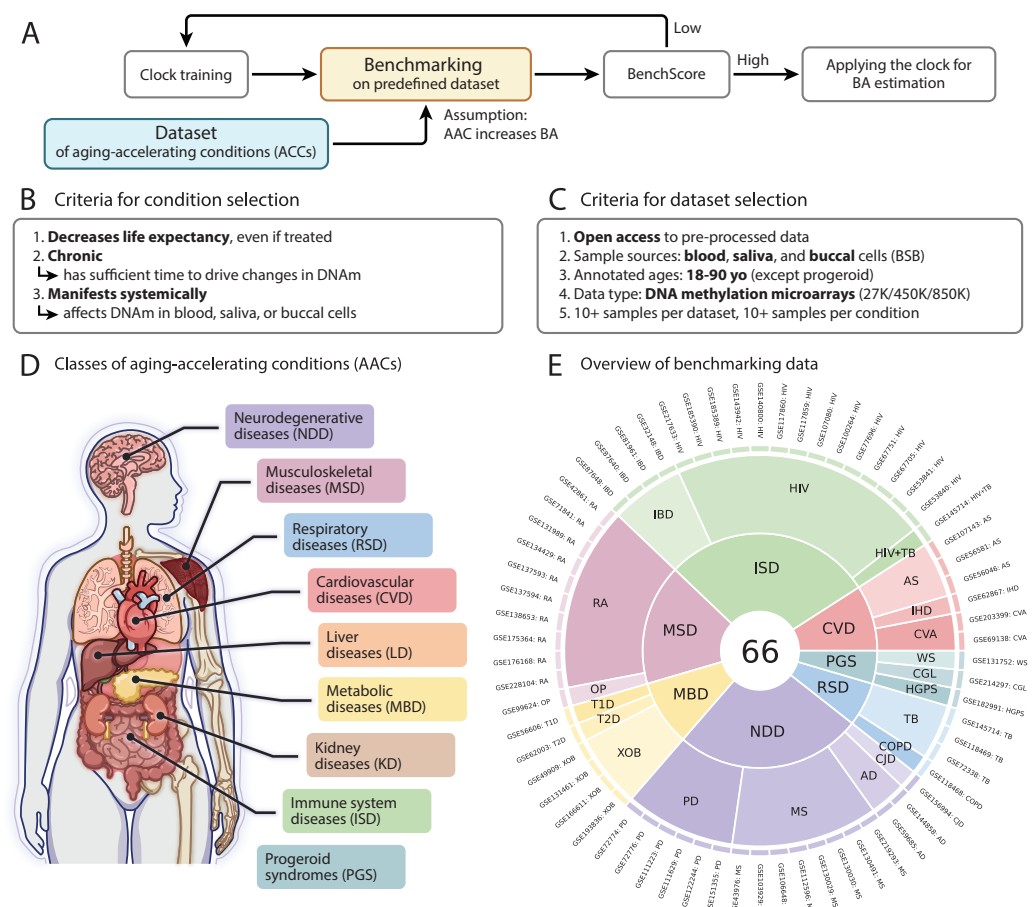

Figure 2: ComputAgeBench methodology. A) The proposed pipeline for constructing aging clocks features an important step of validating the model on pre-defined aging-acceleration conditions that satisfy criteria (B) and are collected into datasets that meet criteria (C) for individual study design. D) Major classes that include putative aging-accelerating conditions. E) Aggregated dataset panel for benchmarking aging clocks, comprising 66 unique data sources (labeled by their Gene Expression Omnibus dataset identification numbers and conditions) from more than 50 studies. See Table A2 for the full names and Table A3 for the population-based evidence for including each condition.

## 3.3 COLLECTING AAC DATASETS FOR BENCHMARKING

To cover as many organismal systems affected by age-related conditions as possible, we split the aggregated data into nine broad categories (Fig. 2D): cardiovascular diseases (CVD), immune system diseases (ISD), kidney diseases (KDD), liver diseases (LVD), metabolic diseases (MBD), musculoskeletal diseases (MSD), neurodegenerative diseases (NDD), respiratory diseases (RSD), and progeroid syndromes (PGS). In each class, we identified several AACs relying on the established lists of age-related diseases and on the leading causes of death (Mei et al., 2023; Li et al., 2021; Ferrari et al., 2024), including closely associated conditions and other conditions mentioned in a variety of epigenetic clock studies (Horvath, 2013; Levine et al., 2018; Ying et al., 2024; Mei et al., 2023; Horvath et al., 2018). The corresponding AACs with their abbreviations and population-based evidence for their inclusion are provided in Appendix (Tables A2 and A3, respectively).

Dataset search was performed using the NCBI Gene Expression Omnibus (GEO) database, an *omics* data repository with unrestricted access (https://www.ncbi.nlm.nih.gov/geo/). We applied filters to include the *Homo sapiens* species and all types of methylation-related studies: methylation profiling by single-nucleotide polymorphism (SNP) array, methylation profiling by

array, methylation profiling by genome tiling array, and methylation profiling by high throughput sequencing (methylation microarray data can be found in any of these study types).

Upon performing the dataset search, only a portion of AACs from seven condition classes were retained (see Appendix and Table A3). All five dataset selection criteria were met by none of the found kidney- and liver-related AAC datasets. The resulting list of 66 datasets (Reynolds et al., 2014; Nazarenko et al., 2015; Soriano-Tárraga et al., 2016; Istas et al., 2017; Cullell et al., 2022; Harris et al., 2012; Horvath & Levine, 2015; Gross et al., 2016; Zhang et al., 2016; Li Yim et al., 2016; Ventham et al., 2016; Zhang et al., 2017; 2018; Oriol-Tordera et al., 2020; DiNardo et al., 2020; Oriol-Tordera et al., 2022; Esteban-Cantos et al., 2023; Liu et al., 2013; Fernandez-Rebollo et al., 2018; Rhead et al., 2017; Clark et al., 2020; Tao et al., 2021; de la Calle-Fabregat et al., 2021; Julià et al., 2022; Chen et al., 2023; Day et al., 2013; Rakyan et al., 2011; Lunnon et al., 2015; Ramos-Molina et al., 2019; Noronha et al., 2022; Marabita et al., 2013; Lunnon et al., 2014; Horvath & Ritz, 2015; Castro et al., 2019; Kular et al., 2018; Chuang et al., 2017; 2019; Ntranos et al., 2019; Ewing et al., 2019; Carlström et al., 2019; Roubroeks et al., 2020; Go et al., 2020; Dabin et al., 2020; Bingen et al., 2022; Esterhuyse et al., 2015; Chen et al., 2021; 2020; Maierhofer et al., 2019; Bejaoui et al., 2022; Qannan et al., 2023) comprises 65 blood studies and 1 saliva study, and is visualized in Fig. 2E. An overview of all datasets, dataset sizes, and their age distributions is provided in Fig. A1. Descriptive statistics for all datasets are provided in Fig. A2.

We unified the metadata of all datasets by retrieving only the relevant metadata columns and formatting them into the appropriate data types, similarly to what was proposed by the authors of Biolearn (Ying et al., 2023), another recent effort in the clock community. We also added the condition and condition class annotation, thus obtaining a single metadata file with 10,410 rows (samples) and the following columns: SampleID, DatasetID (dataset GEO accession number), PlatformID (sequencing platform), Tissue (blood or saliva), CellType (whole blood or cell type after sorting), Gender, Age, Condition, and Class (see also Appendix A.9 for details on data processing).

### 3.4 EPIGENETIC AGE PREDICTORS

Any blood-based epigenetic aging clock that predicts BA in age units (or can be re-scaled to them) can be validated in our benchmark. We tested 13 publicly available epigenetic clock models trained on adult human data to evaluate sample age (Table A4), with the model coefficients retrieved from the corresponding studies. Among the collected first-generation clocks, 6 were trained purely on blood samples (Hannum et al., 2013; Ying et al., 2024; Lin et al., 2016; Vidal-Bralo et al., 2016), and 3 models were trained on multiple tissues (Horvath, 2013; Zhang et al., 2019; Horvath et al., 2018). Among the second-generation clocks, all were blood-based, and 2 models relied entirely on CpG sites as predictive features (Levine et al., 2018; Higgins-Chen et al., 2022), while the other 2 required additional information about gender and chronological age as inputs (Lu et al., 2019; 2022). Because the extracted datasets contained missing values, we imputed them with the "gold standard" beta values averaged for each CpG site retrieving them from the R "SeSAMe" package (Zhou et al., 2018) (for the results on comparing imputation methods, see Appendix A.3). We also ensured that no data in the benchmark was used to train any of the selected clocks, and that all clock input and output structures are consistent with each other ("harmonized", as described by Ying et al. (2023)). The clock models evaluated by us are described in Table A4.

### 3.5 BENCHMARKING TASKS FOR EVALUATING AGING CLOCKS

To benchmark aging clock models, we propose four tasks: relative aging acceleration prediction (Fig. 3A), absolute aging acceleration prediction (Fig. 3B), chronological age prediction accuracy (Fig. 3C), and systematic chronological age prediction bias (Fig. 3D). In the first two tasks, the clocks are tested if they can correctly predict aging acceleration in the predefined panel of AAC datasets.

In the relative aging acceleration prediction task (AA2 task), we test aging clock ability to distinguish AAC from healthy control (HC) samples in a dataset containing both sample groups. After predicting ages in each dataset corresponding to this task using various clock models, we apply a two-sample Welch's test per dataset and calculate a one-sided P-value (*i.e.*, $H_A : \Delta_{AAC} > \Delta_{HC}$) to determine if mean aging acceleration in the AAC cohort is significantly greater than that in the HC cohort (Fig. 3A). Next, we apply the Benjamini-Hochberg correction procedure for controlling the

false discovery rate (FDR) of predictions across all datasets, with an adjusted P-value less than 0.05 considered indicative of statistical significance. We selected a parametric test due to the assumption of normal distribution of $\Delta$, a fundamental trait of the multivariate linear regression models commonly used in aging clock construction.

In the absolute aging acceleration prediction task (AA1 task), we test clock ability to correctly predict positive aging acceleration for an AAC in the absence of the HC cohort. For each dataset in this task, we predict ages using various clock models, apply a one-sample Student's t-test and calculate a one-sided P-value (*i.e.*, $H_A : \Delta_{AAC} > 0$) to determine if mean aging acceleration in the AAC cohort is significantly greater than zero (Fig. 3B). As before, we apply the Benjamini-Hochberg correction procedure for controlling FDR with the same adjusted P-value threshold.

Clearly, the first task (AA2) provides a more rigorous way to test aging clocks compared to AA1, because it helps to control potential covariate shifts, but the second task (AA1) deserves its place in the list, as it allows including more data into the panel to overcome data scarcity. The third task is aimed at distinguishing good predictors of chronological age from predictors of biological age. Due to the paradox of biomarkers mentioned above, it is highly unlikely that the same model could combine both these properties. Yet, the good predictors of chronological age are believed to be useful in forensics (Paparazzo et al., 2023) or data labeling, where the chronological age information is lacking. We chose median absolute aging acceleration ($Med(|\Delta|)$), a full equivalent of median absolute error, for testing clock performance. We calculate it across HC samples from the whole dataset panel and report it as a single number expressed in years.

We introduced the fourth task, a prediction bias task, to evaluate the robustness of a given aging clock model to covariate shift between the original clock training dataset and the datasets from the proposed benchmark. Covariate shift, also referred to as batch effect in bioinformatics, denotes the shift between covariate distributions in two datasets. For instance, the distribution of methylation values for a given CpG site could be centered around 0.45 in one dataset and around 0.55 in the other one—a common scenario in DNAm and other omics data. Because each clock is trained on healthy controls, we expect age deviation of HC samples to be zero on average (*i.e.*, $E(\Delta_{HC}) = 0$). In practice, however, due to the presence of a covariate shift between the training and testing data, a clock might produce biased predictions, resulting in a systemic bias and adding or subtracting extra years for a healthy individual coming from an external dataset. The goal of the fourth task is to control for such systemic bias in clock predictions. Therefore, as a benchmarking metric for this task, we calculated median aging acceleration ($Med(\Delta)$) across HC samples from the entire dataset panel, which reflects the systematic shift in clock predictions caused by differences between datasets.

## 3.6 Cumulative benchmarking score

We define cumulative benchmarking score such that it would account for the main drawback of AA1 task, namely, the sensitivity to positive model bias. Let $S_{AA2}$ denote total score of a model in AA2 task and $S_{AA1}$ from the AA1 task (both $S_{AA2}$ and $S_{AA1}$ represent the number of datasets evaluated correctly by a model in the respective task), then the cumulative benchmarking score is:

$$BenchScore = S_{AA2} + S_{AA1} \cdot \left(1 - \frac{\max(0, Med(\Delta))}{Med(|\Delta|)}\right). \tag{2}$$

Consequently, if a model is positively biased, its performance in the AA1 task will be penalized by the bracketed coefficient by the $S_{AA1}$, the largest when the model bias $Med(\Delta)$ is zero. Because $Med(\Delta) \leq Med(|\Delta|)$, this coefficient is limited to the $[0, 1]$ interval.

While designing our metric, we aimed for simplicity and interpretability. At the same time, we sought to include more data in the benchmark to address data scarcity caused by the underrepresentation of certain AACs. Admittedly, there could be a more optimal solution for the metric, but we also believe that such a solution must be proposed by a continuous collaborative discussion between the aging clock and machine learning communities, which we are eager to establish.

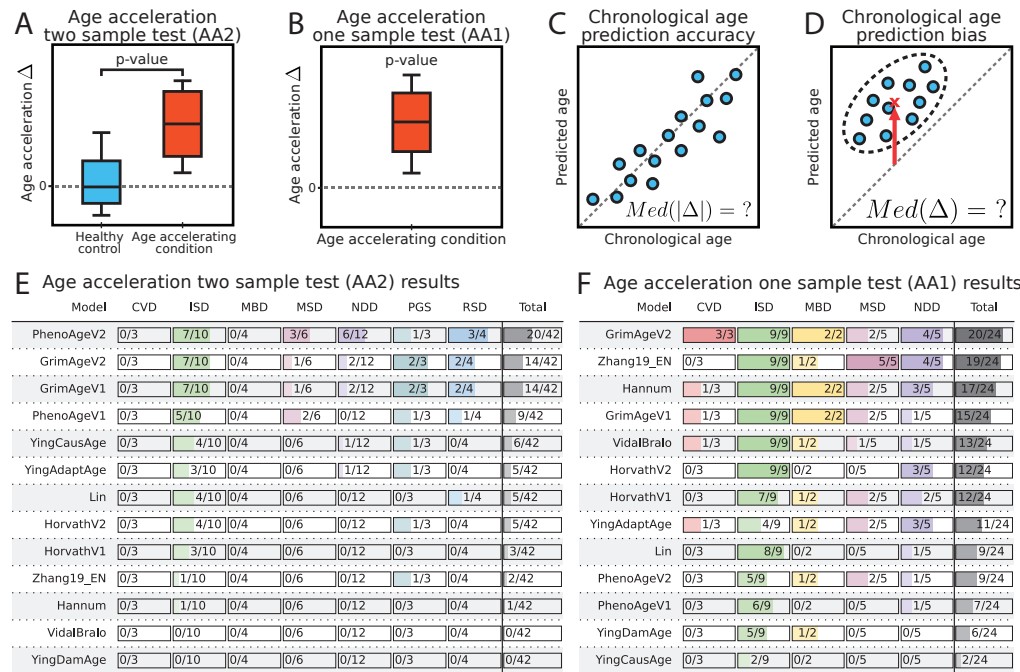

Figure 3: ComputAgeBench tasks and performance of aging clock models. A-D) The four benchmarking tasks. (C) illustrates that chronological age prediction accuracy is measured by median absolute error ($Med(|\Delta|)$) across all predictions. For a limiting case of prediction bias sketched in (D), all samples were predicted with positive age acceleration, leading to a strictly positive value of $Med(\Delta)$, graphically represented as a red arrow pointing to a cross. E) AA2 task results split into columns by condition class, where scores demonstrate the number of datasets per class, where a given clock model detected significant difference between the HC and AAC cohorts. F) AA1 task results: same as (E), but the statistics are calculated for datasets containing the AAC cohort alone.

## 4 RESULTS

The most rigorous of the four, AA2 task demonstrates that second-generation aging clocks (PhenoAgeV2 (Higgins-Chen et al., 2022), GrimAgeV1 (Lu et al., 2019), GrimAgeV2 (Lu et al., 2022), and PhenoAgeV1 (Levine et al., 2018)) appear on top, particularly at predicting aging acceleration for the ISD class (Fig. 3E, Supplementary Materials Fig. A5). Nevertheless, all clocks failed to detect aging acceleration in patients with cardiovascular and metabolic diseases, at least at the statistically significant level (see Figs. A3 and A4 for results without FDR correction). Modest scores (<50% datasets in total) on the AA2 task across all models are expected, as no clocks had specifically been calibrated to pass this benchmarking task.

In contrast, the first-generation aging clocks by Zhang et al. (2019) and Hannum et al. (2013) populated the top of the AA1 leaderboard, in addition to the GrimAge, exhibiting good scores across multiple condition classes (Fig. 3F, Supplementary Materials Fig. A6). Notably, combining the results of this task with the model bias task exposes the potential source of the exceptional "robustness" in predicting accelerated aging in datasets without healthy controls.

The task of chronological age prediction accuracy reveals two undeniable leaders: HorvathV2 (Horvath et al., 2018) and HorvathV1 (Horvath, 2013) clocks (Table 1), specifically tuned for this task on large multi-tissue datasets. Notably, clocks predicting chronological age with $Med(|\Delta|) \geq 18$ years would be inferior to a constant model yielding a 50 y.o. prediction (average age across all HC samples in the benchmark). Unless scaled, such clocks can hardly be used for inferring accelerated aging.

Finally, to prove the validity of AA1 performance, a clock should also pass the task for being unbiased. We show that the AA1 leader, GrimAgeV2 clock (Lu et al., 2022), is also characterized by

Table 1: Benchmarking results.

| Model name | AA2 score | AA1 score | $Med(|\Delta|)$, years | $Med(\Delta)$, years | $BenchScore$ |
|---|---|---|---|---|---|
| **PhenoAgeV2** | **20** | 9 | 7.6 ±0.1 | -2.6 ±0.1 | **29.0** |
| GrimAgeV1 | 14 | 15 | 7.5 ±0.1 | 5.7 ±0.1 | 17.4 |
| PhenoAgeV1 | 9 | 7 | 8.0 ±0.1 | -4.2 ±0.2 | 16.0 |
| GrimAgeV2 | 14 | 20 | 9.8 ±0.1 | 9.3 ±0.1 | 15.1 |
| HorvathV1 | 3 | 12 | 5.4 ±0.1 | -0.1 ±0.1 | 15.0 |
| HorvathV2 | 5 | 12 | **4.1 ±0.1** | 1.1 ±0.1 | 13.9 |
| VidalBralo | 0 | 13 | 9.1 ±0.1 | 0.1 ±0.2 | 12.8 |
| Lin | 5 | 9 | 7.5 ±0.1 | 2.1 ±0.2 | 11.4 |
| YingAdaptAge | 5 | 11 | 20.0 ±0.2 | 12.5 ±0.5 | 9.1 |
| YingCausAge | 6 | 2 | 9.0 ±0.1 | 1.3 ±0.2 | 7.7 |
| YingDamAge | 0 | 6 | 19.5 ±0.3 | -14.5 ±0.5 | 6.0 |
| Zhang19_EN | 2 | 19 | 10.5 ±0.2 | 9.6 ±0.2 | 3.7 |
| Hannum | 1 | 17 | 7.5 ±0.1 | 6.3 ±0.1 | 3.7 |

a large prediction bias for the HC samples (Table 1), warning us against considering its AA1 task score reliable. On the other hand, the top-2 unbiased HorvathV1 clock (Horvath, 2013) and Vidal-Bralo clock (Vidal-Bralo et al., 2016) have low prediction bias, rendering their AA1 performance as more trustworthy.

To account for the discrepancies of AA1 task interpretation regarding the prediction bias, we devised *cumulative benchmarking score* (Table 1) which penalizes AA1 score by the magnitude of prediction bias (see Eq. 2). With such a metric, a second-generation aging clock PhenoAgeV2 (Higgins-Chen et al., 2022) becomes the most robust model in terms of distinguishing individuals with aging-accelerating conditions from the healthy cohort. This model is a leader, according to the cumulative benchmarking score and the AA2 task score. Closely behind it, are the other second-generation clocks: GrimAgeV1 (Lu et al., 2019), PhenoAgeV1 (Levine et al., 2018), and GrimAgeV2 (Lu et al., 2022). On the other hand, our results indicate that even the classic first-generation aging clocks, such as HorvathV1 (Horvath, 2013) and HorvathV2 (Horvath et al., 2018), can perform quite reliably in predicting biological age, at least for some condition classes. It is noteworthy that in both AA1 and AA2 tasks, many aging clocks perform well in detecting accelerated aging caused by immune system diseases, which are mostly represented by human immunodeficiency virus (HIV) infection in our dataset, while the other disease classes are only captured by *some* clocks, allegedly indicating that they were implicitly and unintentionally trained for certain subset of diseases. These results generalize previous findings (Mei et al., 2023) and show that comprehensive benchmarking of aging clocks can resolve the controversy regarding their robustness and utility.

## 5 DISCUSSION

Biological age is an elusive concept that cannot be measured and validated directly, which necessitates careful choice of model assumptions to avoid methodological errors and false discoveries while estimating it. While maintaining some degree of correlation between predicted and chronological age is desirable, the biomarkers paradox (Klemera & Doubal, 2006) precludes one from automatically accepting a BA estimation as acceptable (via the classic performance metrics of chronological age prediction accuracy). From a methodological perspective, training BA predictors to estimate time to death or a disease onset remains the most rigorous approach to aging clock validation, as these events can be measured directly. However, obtaining such data is challenging due to various ethical and financial constraints. At present, no open access data of DNA methylation with mortality labels are available for public clock benchmarking (see Appendix A.7).

While mortality data remain unavailable, we propose to validate clocks by their ability to demonstrate BA acceleration *in a fixed pre-determined panel of datasets* for established aging-accelerating diseases or predict decelerated aging in the datasets of lifespan-prolonging interventions. For that, we developed our benchmark, where each aging clock could be tested across 4 distinct tasks. **We gathered an unprecedented number of DNA methylation datasets from more than 50 studies,**

**covering 19 putative aging-accelerating conditions.** Notably, no aging-decelerating conditions have been confirmed for the benchmark study (see Appendix A.5). It should be taken into account that *in vitro* cell reprogramming cannot serve as validation data for the deceleration effect, because, as has previously been shown (Kriukov et al., 2023), such data are essentially out-of-domain with regard to blood DNA methylation across aging.

To showcase our benchmark, we tested 13 different published models and revealed that the second-generation aging clocks, namely, PhenoAge (Levine et al., 2018), GrimAge (Lu et al., 2019), and their upgraded variants (Higgins-Chen et al., 2022; Lu et al., 2022), were the most successful, according to the cumulative benchmarking score. As these clocks had initially been designed to predict all-cause mortality, they were expected to be robust in distinguishing aging-accelerating conditions. Yet, our findings reinforce the growing trends in training BA predictors based on mortality rather than chronological age (Yousefi et al., 2022; Moqri et al., 2024).

As blood DNA methylation generally comes from the immune cells, which would be directly affected by the HIV, it is not surprising that the majority of clocks managed to discern accelerated aging in the immune system-related conditions (featured predominantly by the HIV infection in our dataset). This result supports the notion that the blood-based clocks might be implicitly attuned to such conditions, while only a few clocks are capable of successfully capturing accelerated aging in the other disease classes.

Remarkably, some datasets were evaluated incorrectly *by all models*, which may have several possible explanations apart from the poor clock performance. First, a strong covariate shift between these data and the training data might impede model performance on some datasets. Second, some selected conditions might not induce accelerated aging in blood, either by itself or by the design of the original study (see Limitations in Appendix A.1). Third, the multidimensionality of aging as a biological phenomenon might not allow for correct prediction of all aging-accelerating conditions by such univariate measures as the blood-based epigenetic clocks. In favor of this notion, it has recently been shown that different organ systems have different aging trajectories (Schaum et al., 2020; Oh et al., 2023), suggesting several directions for the future research, outlined in Appendix A.2.

## 6 CONCLUSION

In this work, we developed the first systematic benchmark for evaluating blood-based epigenetic aging clocks. We believe it will help longevity researchers and data scientists to better gauge the power of existing biomarkers of aging, quantitatively assessing their role, limitations, and reliability. We anticipate that, as a result of such computational paradigm, rapid and reliable clinical trials of lifespan-extending therapies will become an attainable reality in a not-so-distant future.

## 7 REPRODUCIBILITY STATEMENT

We assured the reproducibility of our pipeline by providing a Google Colab notebook (https://colab.research.google.com/drive/1_nrGMUd8oH8ADNWUPNeXHr4ZAJlZOQhm), which allows to download all datasets and benchmark all clocks considered in this article. References to our code and dataset repositories will become available after the double-blind review.

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

# A   APPENDIX

## A.1   LIMITATIONS

The current version of our benchmark harbors several important limitations. First, some selected conditions might not actually fulfill the suggested criteria, especially regarding their effect on blood DNA methylation, although we did our best to identify the most unambiguous ones. From the other hand, some conditions that fit our criteria might have escaped our attention. Second, the conditions are not represented uniformly, with some being featured in 10+ datasets (HIV, rheumatoid arthritis), and some present in a single dataset with few samples (ischemic heart disease, chronic obstructive pulmonary disease, congenital generalized lipodystrophy). The third limitation arises from the known issue of hidden subgroups of patients and mislabeled instances (Varoquaux & Cheplygina, 2022). For the AAC cohorts, having hidden co-morbidities is acceptable, as they would supposedly exaggerate aging acceleration even stronger. Conversely, having severe, but unlabeled diseases in the HC cohort would likely substantially alter the findings of our benchmark. Unfortunately, we can neither expand our dataset to cover all conditions equally, nor explicitly confirm if all studies at hand comply with our requirements.

## A.2   FUTURE WORK

We plan to further extend our benchmarking dataset by incorporating open access data of additional modalities, such as clinical biochemistry, transcriptomics, proteomics, metabolomics, etc. To overcome the aforementioned limitations, we strongly urge an open discussion on developing a panel of conditions and datasets that would serve as the gold standard for reliable and comprehensive validation of emerging biomarkers of aging. We also believe that it is important to expand the benchmark to animal models, since collecting the required data and developing preclinical biomarkers of aging in some animals is associated with fewer ethical and financial challenges. Hopefully, all these issues and developments will be addressed by the efforts of a recently established Biomarkers of Aging Consortium (https://www.agingconsortium.org/). Ultimately, the "correct" BA estimator should satisfy all four properties we defined in the Introduction. Regardless of the clock generation or data modality, reliable aging clock models must also be able to assess the uncertainty of their own predictions before being integrated in clinical trials (Chua et al., 2023; Kriukov et al., 2023). And indeed, an example of uncertainty-aware aging clocks has recently been proposed (Varshavsky et al., 2023). We also aim to upgrade our package to facilitate the interaction with other clock-related resources, including Biolearn (Ying et al., 2023) and pyaging (de Lima Camillo, 2024).

## A.3   COMPARISON OF DIFFERENT APPROACHES TO MISSING VALUES IMPUTATION

We ran additional experiments (see Table A1) to test different imputation methods and observed that the method we used (Sesame450k) leads to the most accurate age predictions across all models except the VidalBralo clock, whose MAE is $0.19\%$ lower when using imputation by zeros. We did not have to impute all 800k+ sites in the whole dataset, as we only imputed sites included in each respective clock model. Importantly, the results in other benchmarking tasks remained intact, regardless of the imputation strategies.

## A.4   SOCIETAL IMPACT

The obvious positive societal impact of our work is the prospect of increased active lifespan and that of healthy longevity. Our benchmarking methodology assists in determining the most accurate predictors of the biological age, which, in turn, assists in delineating the crucial biomarkers and factors that might prolong the healthy life. The potential negative impact entails the common issues emphasized when a fundamental biological problem is tackled with the AI tools. Specific to the subject of longevity are the issues of pre-mature excitement in the mass media when a certain factor is hypothesized to prolong life. A relevant fraud in the pharmaceutical industry is also plausible, if not regulated. One could also envision the depletion of resources caused by an overpopulation of the Earth, which might happen if the longevity drug is found. These negative possibilities are not expected to be sudden and could be mitigated gradually – similarly to a plethora of other benchmarking works established for solving important biological problems.

Table A1: MAE results (in years) for different strategies of missing values imputation.

| Model name | Sesame 450k | Average | Zeros |
|---|---|---|---|
| HorvathV2 | **4.143847** | 4.701762 | 4.719477 |
| HorvathV1 | **5.350622** | 5.475857 | 5.475857 |
| GrimAgeV1 | **7.462245** | 8.102066 | 8.241653 |
| Lin | **7.467559** | 8.367630 | 8.429655 |
| Hannum | **7.477633** | 7.890421 | 7.907489 |
| PhenoAgeV2 | **7.604413** | 8.439977 | 8.432397 |
| PhenoAgeV1 | **8.009677** | 8.380239 | 8.381561 |
| YingCausAge | **8.969959** | 11.599078 | 11.551690 |
| VidalBralo | 9.124225 | 9.124387 | **9.107015** |
| GrimAgeV2 | **9.796544** | 10.513198 | 10.576180 |
| Zhang19_EN | **10.534452** | 10.611938 | 10.611938 |
| YingDamAge | **19.534224** | 20.179561 | 20.211066 |
| YingAdaptAge | **19.972273** | 23.287544 | 23.353844 |

## A.5 MOTIVATION BEHIND INCLUDING OR EXCLUDING PARTICULAR CONDITIONS

Our first criterion for selecting aging-accelerating conditions (AACs) was that having an AAC must lead to decreased life expectancy (LE) compared to the general population, even when treated with existing therapies. As we have mentioned earlier, this decrease in LE and the corresponding increase in mortality must result mainly from intrinsic organismal causes rather than from socioeconomic factors and self-destructive behaviors related to a given condition. Thus, while Down syndrome (DS) is associated with elevated prevalence of multiple chronic diseases (O'Leary et al., 2018; Landes et al., 2020; Baksh et al., 2023), LE of DS individuals has grown dramatically by over 450% from 1960 to 2007 (Presson et al., 2013), even though no cure for DS has been developed, suggesting strong non-biological confounding factors at play. Additionally, while some authors expect DS to display accelerated epigenetic aging (Horvath, 2013), others anticipate deceleration when applying epigenetic clocks to DS blood samples, as DS individuals are hypothesized to feature juvenile blood (Mei et al., 2023). Schizophrenia (SZ) is another example of a controversial condition: while we can find increased incidence of age-related comorbidities such as cardiovascular diseases, cancers, or chronic obstructive pulmonary disease (Olfson et al., 2015; Oakley et al., 2018; Yung et al., 2021), the rates of suicide and substance-induced death are also increased in people with SZ (Olfson et al., 2015). We therefore suggest excluding such ambiguous conditions from robust clock benchmarking, as it is currently difficult to disentangle functional organismal deterioration from external and behavioral condition-related confounders and evaluate the degree to which the latter influence LE.

Regarding cancers in general, it is difficult to formulate a pre-hoc hypothesis about the directionality of epigenetic age changes. Even though we know that DNAm can be used to create signatures of various cancers, and that changes in some DNAm sites are shared between aging and cancers (Yu et al., 2020), we cannot be certain that an aging clock would indicate accelerated aging in cancerous samples, as some cancer-specific and stem cell-like features such as telomere maintenance might prompt a clock model to treat it as a marker of partial rejuvenation. In support of these considerations, epigenetic age predictions were found to exhibit no correlation with multiple TCGA cancer types (Lin & Wagner, 2015). To avoid possible speculation as far as possible, we recommend excluding cancer from clock benchmarking, as it is difficult to hypothesize about clock performance in such complex phenomena.

Aging-decelerating condition (ADC) is defined as a condition that increases LE compared to the general population and features the same second and third criteria as an AAC. With respect to human data, however, the ADCs are difficult to determine, as human lifespan-increasing interventions are yet to emerge. There are genetic mutations, such as Laron syndrome (growth hormone insensitivity) or isolated growth hormone deficiency (growth hormone releasing hormone insensitivity), that appear to protect against some age-related pathologies, but they do not feature a prolonged lifes-

pan (Aguiar-Oliveira & Bartke, 2019). To avoid dubious interpretation, we recommend omitting the inclusion of any condition into the ADC category when benchmarking human aging clocks.

The resulting list of condition classes and conditions selected to represent accelerated aging is listed in Table A2. Population-based evidence for condition inclusion and the number of datasets found and selected per condition are displayed in Table A3.

Table A2: Aging-accelerating conditions. ICD-10: class or condition code(s) from the International Classification of Diseases Version 10; a dash indicates lack of specific code; abbr.: abbreviation.

| Condition class | Class ICD-10 | Class abbr. | Aging-accelerating condition (AAC) | Condition ICD-10 | Condition abbr. |
|---|---|---|---|---|---|
| Cardio-vascular diseases | I00-I99 | CVD | Atherosclerosis | I70 | AS |
| | | | Ischemic heart disease | I20-I25 | IHD |
| | | | Cerebrovascular accident | I60-I63 | CVA |
| | | | Heart failure | I50 | HF |
| | | | Myocardial infarction | I21-I22 | MCI |
| Immune system diseases | — | ISD | Inflammatory bowel disease | K50-K51 | IBD |
| | | | Human immunodeficiency virus infection | B20-B24 | HIV |
| Kidney diseases | N00-N99 | KDD | Chronic kidney disease | N18 | CKD |
| Liver diseases | K70-K77 | LVD | Nonalcoholic steatohepatitis | K75.81 | NASH |
| | | | Primary biliary cholangitis | K74.3 | PBC |
| | | | Primary sclerosing cholangitis | K83.01 | PSC |
| | | | Cirrhosis | K70.3, K74.3-K74.6 | CIR |
| Metabolic diseases | E00-E90 | MBD | Extreme obesity | E66.01, E66.2 | XOB |
| | | | Type 1 diabetes | E10 | T1D |
| | | | Type 2 diabetes | E11 | T2D |
| | | | Metabolic syndrome | E88.810 | MBS |
| Musculo-skeletal diseases | M00-M99 | MSD | Sarcopenia | M62.84 | SP |
| | | | Osteoporosis | M80-M81 | OP |
| | | | Osteoarthritis | M15-M19 | OA |
| | | | Rheumatoid arthritis | M05-M06 | RA |
| Neuro-degenerative diseases | G00-G99 | NDD | Alzheimer's disease | G30 | AD |
| | | | Parkinson's disease | G20 | PD |
| | | | Multiple sclerosis | G35 | MS |
| | | | Dementia with Lewy bodies | G31.83 | DLB |
| | | | Creutzfeldt-Jakob disease | A81.0 | CJD |
| Respiratory diseases | J00-J99 | RSD | Chronic obstructive pulmonary disease | J44 | COPD |
| | | | Idiopathic pulmonary fibrosis | J84.112 | IPF |
| | | | Tuberculosis | A15 | TB |
| Progeroid syndromes | — | PGS | Werner syndrome | E34.8 | WS |
| | | | Hutchinson-Gilford progeria syndrome | E34.8 | HGPS |
| | | | Congenital generalized lipodystrophy | E88.1 | CGL |
| | | | Dyskeratosis congenita | Q82.8 | DKC |

Table A3: Population-based evidence for condition inclusion, and the number of datasets found and selected for each condition. GEO: Gene Expression Omnibus database; abbr.: abbreviation.

| Class abbr. | Condition abbr. | Evidence of decreased life expectancy | N items in the GEO query | N datasets after filtering |
|---|---|---|---|---|
| CVD | AS | Chen et al. (2023a); Costa et al. (2021); Ikeda & Ohishi (2019); Lernfelt et al. (2002); Sutton-Tyrrell et al. (1995) | 22 | 3 |
| | IHD | Martin et al. (2024); Dai et al. (2022); Hartley et al. (2016); Bertuccio et al. (2011) | 21 | 1 |
| | CVA | Martin et al. (2024); GBD 2019 Stroke Collaborators (2021); Xian et al. (2012); Grysiewicz et al. (2008) | 10 | 2 |
| | HF | Martin et al. (2024); Bytyçi & Bajraktari (2015); Shahar et al. (2004) | 14 | 0 |
| | MCI | Martin et al. (2024); Bucholz et al. (2015); Saaby et al. (2014) | 19 | 0 |
| ISD | IBD | Duricova et al. (2010); Canavan et al. (2007); Dong et al. (2020); Gyde et al. (1982); Kuenzig et al. (2020); Selinger & Leong (2012); Card et al. (2003) | 30 | 4 |
| | HIV | Martin et al. (2024); Trickey et al. (2023); Legarth et al. (2016); May et al. (2014); Nakagawa et al. (2013) | 44 | 15 |
| KDD | CKD | Ke et al. (2022); Tonelli et al. (2006); Kim et al. (2019) | 6 | 0 |
| LVD | NASH | Sheka et al. (2020); Younossi et al. (2019) | 8 | 0 |
| | PBC | Sayiner et al. (2019); Lleo et al. (2016) | 1 | 0 |
| | PSC | Card et al. (2008); Kornfeld et al. (1997) | 2 | 0 |
| | CIR | Martin et al. (2024); Xiao et al. (2023); Dam Fialla et al. (2012) | 68 | 0 |
| MBD | XOB | Martin et al. (2024); Kitahara et al. (2014); Masters et al. (2013); Fontaine et al. (2003); Solomon & Manson (1997) | 96 | 4 |
| | T1D | Ruiz et al. (2022); Heald et al. (2020); Rawshani et al. (2018); Huo et al. (2016); Livingstone et al. (2015); Harjutsalo et al. (2011) | 14 | 1 |
| | T2D | Martin et al. (2024); Emerging Risk Factors Collaboration (2023); Zhu et al. (2022b); Wright et al. (2017); Mulnier et al. (2006); Zhu et al. (2022a) | 45 | 1 |
| | MBS | Martin et al. (2024); Käräjämäki et al. (2022); Wu et al. (2010); Mozaffarian et al. (2008) | 17 | 0 |

Table A3: Population-based evidence for condition inclusion, and the number of datasets found and selected for each condition. GEO: Gene Expression Omnibus database; abbr.: abbreviation. (Continued)

| Class abbr. | Condition abbr. | Evidence of decreased life expectancy | N items in the GEO query | N datasets after filtering |
|---|---|---|---|---|
| MSD | SP | Xu et al. (2022); Brown et al. (2016); Chang & Lin (2016) | 2 | 0 |
| | OP | Rashki Kemmak et al. (2020); Abrahamsen et al. (2015); Center et al. (1999); Cherny et al. (2010) | 5 | 1 |
| | OA | Martin et al. (2024); Liu et al. (2022); Fu et al. (2022); Liu et al. (2015) | 26 | 0 |
| | RA | Chiu et al. (2021); Zhang et al. (2017b); Lassere et al. (2013); Jacobsson et al. (1993) | 37 | 10 |
| NDD | AD | Martin et al. (2024); Li et al. (2022); Liang et al. (2021a); Ganguli et al. (2005); Dodge et al. (2003) | 43 | 2 |
| | PD | Macleod et al. (2014); Willis et al. (2012); Posada et al. (2011) | 37 | 6 |
| | MS | Qian et al. (2023); Lunde et al. (2017); Leray et al. (2015) | 29 | 8 |
| | DLB | Liang et al. (2021b); Mueller et al. (2019); Price et al. (2017) | 5 | 0 |
| | CJD | Nishimura et al. (2020); Llorens et al. (2020); Gelpi et al. (2008) | 1 | 1 |
| RSD | COPD | Martin et al. (2024); Park et al. (2019); Ruvuna & Sood (2020); Lange et al. (2016) | 14 | 1 |
| | IPF | Lancaster et al. (2022); Hutchinson et al. (2014); Kolb & Collard (2014); Fernández Pérez et al. (2010) | 14 | 0 |
| | TB | Martin et al. (2024); Menzies et al. (2021); Lee-Rodriguez et al. (2020) | 13 | 3 |
| PGS | WS | Schnabel et al. (2021); Oshima & Hisama (2014); Goto (1997) | 7 | 1 |
| | HGPS | Schnabel et al. (2021); Hennekam (2006) | 14 | 1 |
| | CGL | Lima et al. (2018); Seip & Trygstad (1996) | 1 | 1 |
| | DKC | Al Nuaimi et al. (2020) | 2 | 0 |
| | **Total number of datasets** | | **667** | **66** |

## A.6 ON DATA TYPES USED FOR AGING CLOCKS CONSTRUCTION

Multiple data modalities were previously used for aging clocks construction. Some examples beyond DNA methylation data include also clinical blood samples (Putin et al., 2016), psycho-social questionnaires (Zhavoronkov et al., 2020), facial images (Xia et al., 2020), urine metabolites (Hertel et al., 2016), and different omics data, gene expression (Holzscheck et al., 2021), DNA accessibility (Morandini et al., 2024), plasma proteins (Sathyan et al., 2020), etc. Interestingly, DNA methylation data allow one the most accurate prediction of chronological age compared to other data modalities, second only to facial imaging data (Xia et al., 2021), and it continues to be used most widely in aging clock construction (Rutledge et al., 2022). It is also important to note that from a practical point of view, in order to construct a clinically relevant aging clock, the method of obtaining the data

should not be too invasive and heavy-handed. For this reason, many clock developers prefer using blood, saliva, or buccal epithelial samples as data sources.

## A.7 ON ACCESSIBILITY OF EXISTING EPIGENETIC MORTALITY DATA

Although there are some existing biobanks that aggregate sensitive human data and provide them in an open-access manner, (e.g., NHANES: https://wwwn.cdc.gov/nchs/nhanes/), most biobanks rely on authorized access to their data (e.g., UK Biobank: https://www.ukbiobank.ac.uk/). The similar semi-open situation occurs with DNA methylation data. Here, we provide information about 12 cohort studies containing DNA methylation data and mortality/morbidity information simultaneously, but all of which allow downloading their data upon a reasonable request by contacting with the principal investigators of each cohort or by requesting data on a special platform. These studies include the Framingham Heart Study (FHS), the Women's Health Initiative (WHI), the Lothian Birth Cohorts (LBC), the Atherosclerosis Risk in Communities (ARIC), the Cardiovascular Health Study (CHS), the Normative Aging Study (NAS), the Invecchiare in Chianti (InCHIANTi), the Cooperative Health Research in the Region of Augsburg (KORA), the Epidemiologische Studie zu Chancen der Verhütung, Früherkennung und optimierten Therapie chronischer Erkrankungen in der älteren Bevölkerung (ESTHER), the Danish Twin Register sample (DTR), the Rotterdam Study (RS), and the Coronary Artery Risk Development in Young Adults (CARDIA) (Moqri et al., 2024; Huan et al., 2022). While we recognize the risks associated with releasing sensitive patient data into the public domain, we also want to emphasize that comprehensive independent validation of the aging clock is difficult without this important datasets. The confidentiality of this data also does not allow us to use it as part of this open-access benchmark. Instead, we focused on epigenetic data from patients with AACs distributed across human lifespan, which did not contain information on mortality, but was publicly accessible.

## A.8 DNA METHYLATION DATA COLLECTION

As we have mentioned in the Methodology section, dataset search was performed using the NCBI Gene Expression Omnibus (GEO) database, an unrestricted-access omics data repository (https://www.ncbi.nlm.nih.gov/geo/) which shares data using the Open Database License (ODbL). The resulting list of 66 AAC datasets (Reynolds et al., 2014; Nazarenko et al., 2015; Soriano-Tárraga et al., 2016; Istas et al., 2017; Cullell et al., 2022; Harris et al., 2012; Horvath & Levine, 2015; Gross et al., 2016; Zhang et al., 2016; Li Yim et al., 2016; Ventham et al., 2016; Zhang et al., 2017a; 2018; Oriol-Tordera et al., 2020; DiNardo et al., 2020; Oriol-Tordera et al., 2022; Esteban-Cantos et al., 2023; Liu et al., 2013; Fernandez-Rebollo et al., 2018; Rhead et al., 2017; Clark et al., 2020; Tao et al., 2021; de la Calle-Fabregat et al., 2021; Julià et al., 2022; Chen et al., 2023b; Day et al., 2013; Rakyan et al., 2011; Lunnon et al., 2015; Ramos-Molina et al., 2019; Noronha et al., 2022; Marabita et al., 2013; Lunnon et al., 2014; Horvath & Ritz, 2015; Castro et al., 2019; Kular et al., 2018; Chuang et al., 2017; 2019; Ntranos et al., 2019; Ewing et al., 2019; Carlström et al., 2019; Roubroeks et al., 2020; Go et al., 2020; Dabin et al., 2020; Bingen et al., 2022; Esterhuyse et al., 2015; Chen et al., 2021; 2020; Maierhofer et al., 2019; Bejaoui et al., 2022; Qannan et al., 2023) indicated in Table A3 is visualized in Fig. 2E and includes: atherosclerosis (AS), ischemic heart disease (IHD, also known as coronary heart disease), cerebrovascular accident (CVA, also known as stroke), inflammatory bowel disease (IBD, including Crohn's disease and ulcerative colitis), human immunodeficiency virus infection (HIV), extreme obesity (XOB, defined by having BMI $\geq$ 40 kg/m2 (Purnell, 2015; Busebee et al., 2023); also known as class III obesity, severe obesity, or morbid obesity), type 1 diabetes mellitus (T1D), type 2 diabetes mellitus (T2D), rheumatoid arthritis (RA), osteoporosis (OP), Alzheimer's disease (AD), Parkinson's disease (PD), multiple sclerosis (MS), Creutzfeldt-Jakob disease (CJD), chronic obstructive pulmonary disease (COPD), tuberculosis (TB), Werner syndrome (WS, including atypical Werner syndrome), Hutchinson-Gilford progeria syndrome (HGPS, including non-classical progeroid laminopathies), and congenital generalized lipodystrophy (CGL, also known as Berardinelli-Seip lipodystrophy). Age distribution across conditions is demonstrated in Fig. A1. An overview of all datasets and their age distributions is provided in Fig. A2. The information on how patient consent was obtained and which ethics procedures were implemented can be accessed in the respective publications. As per NCBI GEO guidelines, all submitters must "ensure that the submitted information does not compromise participant privacy" (https://www.ncbi.nlm.nih.gov/geo/info/faq.html).

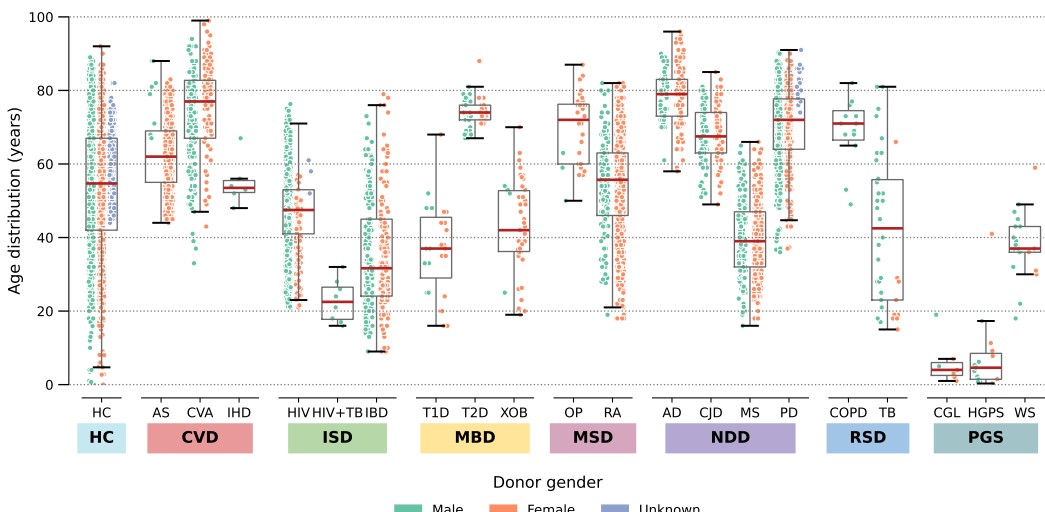

Figure A1: Distribution of dataset samples per condition across ages.

## A.9 DNA METHYLATION DATA PROCESSING

After pre-processing raw output from microarrays or sequencing machines, DNA methylation levels per site are reported quantitatively either as beta values, or as M values. Briefly, beta values represent the ratio of methylated signal (probe intensity or sequencing read counts) to total signal per site (sum of methylated and unmethylated probe intensities or sequencing read counts), while M value is the log2 ratio of the methylated signal versus an unmethylated signal. A more thorough comparison of the two measures can be found in Du et al. (2010). In the original datasets deposited on GEO, DNA methylation values were represented either as a beta fraction (ranging from 0 to 1), beta percentages (ranging from 0 to 100), or M-values (can be both negative and positive, equals 0 when beta equals 0.5). We converted all data to the beta-value fractions ranging from 0 to 1. The values outside this range were treated as missing values (NaNs), as they are not biological. In each dataset, only samples that were relevant for benchmarking (that is, were annotated by age, tissue, and condition) were retained.

The resulting datasets meta-data contains the following fields: DatasetID (datasets GEO ID), PlatformID (GEO ID of a DNA methylation profiling platform), Tissue (sample source tissue: "Blood" stands for peripheral blood samples, "Saliva"—for saliva samples, and "Buccal"—for buccal swab samples), CellType (sample cell type: either a specific cell population, e.g., immune cell subtypes with cell type-specific molecular markers, or broader categories such as whole blood, buffy coat, peripheral blood mononuclear cells (PBMC), or peripheral blood leukocytes (PBL); some samples lack this annotation), Gender (abbreviated sample donor gender: M = Male, F = Female, U = Unknown), Age (sample donor chronological age in years; in the original datasets deposited on GEO, it can be either rounded by the researchers to full years, or converted from months, weeks, or days; where available, we calculated years from the smaller units), Condition (one of AACs or HC), and Class.

As there is no gold standard for DNAm processing, each research group carries out their preferred pipeline that does not necessarily match the processing pipeline used for training the clock model, especially in case of applying earlier clocks (e.g., those by Hannum et al. (2013) or Horvath (2013)). Therefore, so as to retain this typical workflow and not to put any clock model into advantage by choosing the same processing that matches its own pipeline for every dataset, we did not perform any post-processing, inter-dataset normalization, or batch effect correction. In doing so, we also relied on two existing papers. First, compiling already pre-processed datasets without performing the same processing for all of them was done by Ying et al. (2023), another notable effort in the aging clock community. Second, we were also encouraged by a recent work by Varshavsky et al. (2023) who managed to create an accurate clock model by combining several blood datasets—without any additional normalization or correction procedure, using already pre-processed data from previous

| Name | Dataset ID | Platform | Tissue | Class | Condition | N samples | Age distribution |
|------|-----------|----------|--------|-------|-----------|-----------|------------------|
| Reynolds, 2014 | GSE56046 | 450K | B | CVD | AS | 339 / 789 | |
| Reynolds, 2014 | GSE56581 | 450K | B | CVD | AS | 66 / 133 | |
| Nazarenko, 2014 | GSE62867 | 27K | B | CVD | IHD | 6 | |
| Soriano-Tárraga, 2016 | GSE69138 | 450K | B | CVD | CVA | 185 | |
| Istas, 2017 | GSE107143 | 450K | B | CVD | AS | 8 / 8 | |
| Cullell, 2022 | GSE203399 | 450K, 850K | B | CVD | CVA | 121 | |
| Harris, 2012 | GSE32148 | 450K | B | ISD | IBD | 19 / 27 | |
| Horvath, 2015a | GSE53840 | 450K | B | ISD | HIV | 111 | |
| Horvath, 2015a | GSE53841 | 450K | B | ISD | HIV | 24 | |
| Gross, 2016 | GSE67705 | 450K | B | ISD | HIV | 91 / 189 | |
| Horvath, 2015a | GSE67751 | 450K | B | ISD | HIV | 23 / 69 | |
| Zhang, 2016 | GSE77696 | 450K | B | ISD | HIV | 117 / 261 | |
| Li Yim, 2016 | GSE81961 | 450K | B | ISD | IBD | 25 / 15 | |
| Ventham, 2016 | GSE87640 | 450K | B | ISD | IBD | 84 / 156 | |
| Ventham, 2016 | GSE87648 | 450K | B | ISD | IBD | 178 / 204 | |
| Zhang, 2017 | GSE100264 | 450K | B | ISD | HIV | 386 | |
| Zhang, 2017 | GSE107080 | 850K | B | ISD | HIV | 405 | |
| Zhang, 2018 | GSE117859 | 450K | B | ISD | HIV | 608 | |
| Zhang, 2018 | GSE117860 | 850K | B | ISD | HIV | 529 | |
| Oriol-Tordera, 2020 | GSE140800 | 450K | B | ISD | HIV | 70 | |
| | GSE143942 | 450K | B | ISD | HIV | 12 / 42 | |
| DiNardo, 2020 | GSE145714 | 850K | B | ISD | HIV+TB | 8 / 10 | |
| Oriol-Tordera, 2022 | GSE185389 | 450K | B | ISD | HIV | 56 | |
| Oriol-Tordera, 2022 | GSE185390 | 850K | B | ISD | HIV | 30 | |
| Esteban-Cantos, 2023 | GSE217633 | 850K | B | ISD | HIV | 43 / 368 | |
| Day, 2013 | GSE49909 | 27K | B | MBD | XOB | 9 / 40 | |
| Rakyan, 2011 | GSE56606 | 27K | B | MBD | T1D | 29 / 63 | |
| Lunnon, 2015 | GSE62003 | 450K | B | MBD | T2D | 58 | |
| Ramos-Molina, 2019 | GSE131461 | 850K | B | MBD | XOB | 20 | |
| | GSE166611 | 450K | B | MBD | XOB | 17 / 11 | |
| Noronha, 2022 | GSE193836 | 450K | B | MBD | XOB | 11 / 10 | |
| Liu, 2013 | GSE42861 | 450K | B | MSD | RA | 335 / 354 | |
| | GSE71841 | 450K | B | MSD | RA | 12 / 12 | |
| Fernandez-Rebollo, 2018 | GSE99624 | 450K | B | MSD | OP | 16 / 30 | |
| Rhead, 2017 | GSE131989 | 450K | B | MSD | RA | 123 / 230 | |
| Unpublished | GSE134429 | 850K | B | MSD | RA | 17 / 47 | |
| Clark, 2020 | GSE137593 | 850K | B | MSD | RA | 32 | |
| Clark, 2020 | GSE137594 | 850K | B | MSD | RA | 31 | |
| Tao, 2021 | GSE138653 | 850K | B | MSD | RA | 80 | |
| de la Calle-Fabregat, 2021 | GSE175364 | 450K, 850K | B | MSD | RA | 13 / 8 | |
| Julià, 2022 | GSE176168 | 850K | B | MSD | RA | 113 | |
| Chen, 2023 | GSE228104 | 850K | B | MSD | RA | 40 | |
| Marabita, 2013 | GSE43976 | 450K | B | NDD | MS | 52 | |
| Lunnon, 2014 | GSE59685 | 450K | B | NDD | AD | 9 / 48 | |
| Horvath, 2015b | GSE72774 | 450K | B | NDD | PD | 219 / 289 | |
| Horvath, 2015b | GSE72776 | 450K | B | NDD | PD | 38 / 48 | |
| Castro, 2019 | GSE103929 | 450K | B | NDD | MS | 49 | |
| Kular, 2019 | GSE106648 | 450K | B | NDD | MS | 139 / 140 | |
| Chuang, 2017 | GSE111223 | 450K | S | NDD | PD | 131 / 128 | |
| Chuang, 2017 | GSE111629 | 450K | B | NDD | PD | 237 / 335 | |
| Ntranos, 2019 | GSE112596 | 850K | B | NDD | MS | 112 | |
| | GSE122244 | 850K | B | NDD | PD | 34 / 36 | |
| Ewing, 2019 | GSE130029 | 450K | B | NDD | MS | 11 / 20 | |
| Ewing, 2019 | GSE130030 | 450K | B | NDD | MS | 14 / 14 | |
| Carlström, 2019 | GSE130491 | 850K | B | NDD | MS | 82 | |
| Roubroeks, 2020 | GSE144858 | 450K | B | NDD | AD | 96 / 93 | |
| Go, 2020 | GSE151355 | 450K | B | NDD | PD | 20 | |
| Dabin, 2020 | GSE156994 | 450K | B | NDD | CJD | 105 / 114 | |
| Bingen, 2023 | GSE219293 | 850K | B | NDD | MS | 18 / 29 | |
| Esterhuyse, 2015 | GSE72338 | 450K | B | RSD | TB | 20 / 8 | |
| Chen, 2021 | GSE118468 | 450K | B | RSD | COPD | 6 / 15 | |
| Chen, 2020 | GSE118469 | 450K | B | RSD | TB | 9 / 12 | |
| DiNardo, 2020 | GSE145714 | 850K | B | RSD | TB | 10 / 10 | |
| Maierhofer, 2019 | GSE131752 | 850K | B | PGS | WS | 24 / 21 | |
| Bejaoui, 2022 | GSE182991 | 850K | B | PGS | HGPS | 15 / 13 | |
| Qannan, 2022 | GSE214297 | 850K | B | PGS | CGL | 9 | |

Condition classes included in the benchmarking dataset

HC   CVD   ISD   MBD   MSD   NDD   RSD   PGS

Figure A2: Descriptive statistics of datasets included in the benchmark. B: blood, S: saliva. Ages are indicated in years.

Table A4: Aging clock models tested in our benchmark.

| Model name | Number of CpGs | Gene-ration | Extra para-meters | Tissues used for training | Reference |
|---|---|---|---|---|---|
| Hannum | 71 | 1 | — | Blood | Hannum et al. (2013) |
| HorvathV1 | 353 | 1 | — | Multi-tissue | Horvath (2013) |
| Lin | 99 | 1 | — | Blood | Lin et al. (2016) |
| VidalBralo | 8 | 1 | — | Blood | Vidal-Bralo et al. (2016) |
| HorvathV2 | 391 | 1 | — | Blood, Skin | Horvath et al. (2018) |
| PhenoAgeV1 | 513 | 2 | — | Blood | Levine et al. (2018) |
| Zhang19_EN | 514 | 1 | — | Blood, Saliva | Zhang et al. (2019) |
| GrimAgeV1 | 1030 | 2 | Age, Sex | Blood | Lu et al. (2019) |
| GrimAgeV2 | 1030 | 2 | Age, Sex | Blood | Lu et al. (2022) |
| PhenoAgeV2 | 959 | 2 | — | Blood | Higgins-Chen et al. (2022) |
| YingAdaptAge | 999 | 1 | — | Blood | Ying et al. (2024) |
| YingCausAge | 585 | 1 | — | Blood | Ying et al. (2024) |
| YingDamAge | 1089 | 1 | — | Blood | Ying et al. (2024) |

studies (some of which are included in our dataset as well), and thus demonstrating that the between-dataset normalization is not critical for this type of data.

A.10 AGING CLOCKS INCLUDED IN THE BENCHMARKING

The full list of published aging clocks used in this analysis is provided in Table A4.

A.11 BENCHMARKING RESULTS WITHOUT FDR CORRECTION

Figures A3 and A4 demonstrate benchmarking results before applying FDR correction.

| Model | CVD | ISD | MBD | MSD | NDD | PGS | RSD | Total |
|---|---|---|---|---|---|---|---|---|
| PhenoAgeV2 | 0/3 | 7/10 | 0/4 | 4/6 | 6/12 | 1/3 | 3/4 | 21/42 |
| GrimAgeV2 | 0/3 | 7/10 | 0/4 | 3/6 | 2/12 | 2/3 | 3/4 | 17/42 |
| GrimAgeV1 | 0/3 | 7/10 | 0/4 | 3/6 | 2/12 | 2/3 | 2/4 | 16/42 |
| PhenoAgeV1 | 0/3 | 6/10 | 0/4 | 2/6 | 1/12 | 1/3 | 1/4 | 11/42 |
| YingAdaptAge | 0/3 | 6/10 | 0/4 | 1/6 | 1/12 | 2/3 | 0/4 | 10/42 |
| Lin | 0/3 | 7/10 | 0/4 | 0/6 | 1/12 | 0/3 | 1/4 | 9/42 |
| YingCausAge | 0/3 | 4/10 | 0/4 | 1/6 | 1/12 | 2/3 | 0/4 | 8/42 |
| HorvathV2 | 0/3 | 4/10 | 0/4 | 0/6 | 2/12 | 2/3 | 0/4 | 8/42 |
| VidalBralo | 0/3 | 5/10 | 0/4 | 1/6 | 0/12 | 1/3 | 0/4 | 7/42 |
| HorvathV1 | 0/3 | 5/10 | 0/4 | 0/6 | 0/12 | 1/3 | 0/4 | 6/42 |
| Hannum | 0/3 | 3/10 | 0/4 | 1/6 | 1/12 | 1/3 | 0/4 | 6/42 |
| YingDamAge | 0/3 | 3/10 | 0/4 | 0/6 | 1/12 | 1/3 | 0/4 | 5/42 |
| Zhang19_EN | 0/3 | 2/10 | 0/4 | 0/6 | 0/12 | 1/3 | 0/4 | 3/42 |

Figure A3: AA2 task results split into columns by condition class **without FDR correction of P-values**. Scores demonstrate the number of datasets per class, in which a given clock model detected significant (at the 0.05 level of significance) difference between the HC and AAC cohorts.

| Model | CVD | ISD | MBD | MSD | NDD | Total |
|---|---|---|---|---|---|---|
| GrimAgeV2 | 3/3 | 9/9 | 2/2 | 2/5 | 4/5 | 20/24 |
| Zhang19_EN | 0/3 | 9/9 | 1/2 | 5/5 | 4/5 | 19/24 |
| Hannum | 1/3 | 9/9 | 2/2 | 2/5 | 4/5 | 18/24 |
| GrimAgeV1 | 1/3 | 9/9 | 2/2 | 2/5 | 2/5 | 16/24 |
| VidalBralo | 1/3 | 9/9 | 1/2 | 1/5 | 1/5 | 13/24 |
| HorvathV2 | 0/3 | 9/9 | 0/2 | 0/5 | 3/5 | 12/24 |
| HorvathV1 | 0/3 | 7/9 | 1/2 | 2/5 | 2/5 | 12/24 |
| YingAdaptAge | 1/3 | 4/9 | 1/2 | 2/5 | 3/5 | 11/24 |
| PhenoAgeV2 | 0/3 | 6/9 | 1/2 | 2/5 | 1/5 | 10/24 |
| Lin | 0/3 | 8/9 | 0/2 | 0/5 | 1/5 | 9/24 |
| PhenoAgeV1 | 0/3 | 6/9 | 0/2 | 0/5 | 1/5 | 7/24 |
| YingDamAge | 0/3 | 5/9 | 1/2 | 0/5 | 0/5 | 6/24 |
| YingCausAge | 0/3 | 2/9 | 0/2 | 1/5 | 0/5 | 3/24 |

Figure A4: AA1 task results **without FDR correction of P-values**: same as Fig. A3, but the statistics are calculated for datasets containing the AAC cohort only.

## APPENDIX REFERENCES

Bo Abrahamsen, Clive Osmond, and Cyrus Cooper. Life expectancy in patients treated for osteoporosis: Observational cohort study using national danish prescription data. *J. Bone Miner. Res.*, 30(9):1553–1559,

September 2015.

Manuel H Aguiar-Oliveira and Andrzej Bartke. Growth hormone deficiency: health and longevity. *Endocrine reviews*, 40(2):575–601, 2019.

Mohammed Al Nuaimi, Evelyn Elias, Albert Catala, Bozana Zlateska, Yeon Jung Lim, Robert J Klaassen, Geoff DE Cuvelier, Conrad Fernandez, Meera Rayar, MacGregor Steele, et al. Genotypic and phenotypic spectrum of dyskeratosis congenita: Results from the canadian inherited marrow failure registry. *Blood*, 136: 8–9, 2020.

R Asaad Baksh, Sarah E Pape, Li F Chan, Aisha A Aslam, Martin C Gulliford, and Andre Strydom. Multiple morbidity across the lifespan in people with down syndrome or intellectual disabilities: a population-based cohort study using electronic health records. *The Lancet Public Health*, 8(6):e453–e462, 2023.

Yosra Bejaoui, Aleem Razzaq, Noha A Yousri, Junko Oshima, Andre Megarbane, Abeer Qannan, Ramya Potabattula, Tanvir Alam, George M Martin, Henning F Horn, Thomas Haaf, Steve Horvath, and Nady El Hajj. DNA methylation signatures in blood DNA of Hutchinson-Gilford progeria syndrome. *Aging Cell*, 21(2):e13555, February 2022.

Paola Bertuccio, Fabio Levi, Francesca Lucchini, Liliane Chatenoud, Cristina Bosetti, Eva Negri, and Carlo La Vecchia. Coronary heart disease and cerebrovascular disease mortality in young adults: recent trends in europe. *Eur. J. Cardiovasc. Prev. Rehabil.*, 18(4):627–634, August 2011.

Jeremy M Bingen, Lindsay V Clark, Mark R Band, Ilyas Munzir, and Michael D Carrithers. Differential DNA methylation associated with multiple sclerosis and disease modifying treatments in an underrepresented minority population. *Front. Genet.*, 13:1058817, 2022.

Justin C Brown, Michael O Harhay, and Meera N Harhay. Sarcopenia and mortality among a population-based sample of community-dwelling older adults. *J. Cachexia Sarcopenia Muscle*, 7(3):290–298, June 2016.

Emily M Bucholz, Sharon-Lise T Normand, Yun Wang, Shuangge Ma, Haiqun Lin, and Harlan M Krumholz. Life expectancy and years of potential life lost after acute myocardial infarction by sex and race: A cohort-based study of medicare beneficiaries. *J. Am. Coll. Cardiol.*, 66(6):645–655, August 2015.

Bradley Busebee, Wissam Ghusn, Lizeth Cifuentes, and Andres Acosta. Obesity: A review of pathophysiology and classification. In *Mayo Clinic Proceedings*. Elsevier, 2023.

Ibadete Bytyçi and Gani Bajraktari. Mortality in heart failure patients. *Anatol. J. Cardiol.*, 15(1):63–68, January 2015.

C Canavan, K R Abrams, and J F Mayberry. Meta-analysis: mortality in crohn's disease. *Aliment. Pharmacol. Ther.*, 25(8):861–870, April 2007.

Tim Card, Richard Hubbard, and Richard F A Logan. Mortality in inflammatory bowel disease: a population-based cohort study. *Gastroenterology*, 125(6):1583–1590, December 2003.

Tim R Card, Masoud Solaymani-Dodaran, and Joe West. Incidence and mortality of primary sclerosing cholangitis in the UK: a population-based cohort study. *J. Hepatol.*, 48(6):939–944, June 2008.

Karl E Carlström, Ewoud Ewing, Mathias Granqvist, Alexandra Gyllenberg, Shahin Aeinehband, Sara Lind Enoksson, Antonio Checa, Tejaswi V S Badam, Jesse Huang, David Gomez-Cabrero, Mika Gustafsson, Faiez Al Nimer, Craig E Wheelock, Ingrid Kockum, Tomas Olsson, Maja Jagodic, and Fredrik Piehl. Therapeutic efficacy of dimethyl fumarate in relapsing-remitting multiple sclerosis associates with ROS pathway in monocytes. *Nat. Commun.*, 10(1):3081, July 2019.

Kamilah Castro, Achilles Ntranos, Mario Amatruda, Maria Petracca, Peter Kosa, Emily Y Chen, Johannes Morstein, Dirk Trauner, Corey T Watson, Michael A Kiebish, Bibiana Bielekova, Matilde Inglese, Ilana Katz Sand, and Patrizia Casaccia. Body mass index in multiple sclerosis modulates ceramide-induced DNA methylation and disease course. *EBioMedicine*, 43:392–410, May 2019.

J R Center, T V Nguyen, D Schneider, P N Sambrook, and J A Eisman. Mortality after all major types of osteoporotic fracture in men and women: an observational study. *Lancet*, 353(9156):878–882, March 1999.

Shu-Fang Chang and Pei-Ling Lin. Systematic literature review and meta-analysis of the association of sarcopenia with mortality. *Worldviews Evid. Based. Nurs.*, 13(2):153–162, April 2016.

Weihua Chen, Zeya Li, Yu Zhao, Yitian Chen, and Rongchong Huang. Global and national burden of atherosclerosis from 1990 to 2019: trend analysis based on the global burden of disease study 2019. *Chin. Med. J. (Engl.)*, 136(20):2442–2450, October 2023a.

Yulan Chen, Qiao Wang, Haina Liu, Lei Jin, Xin Feng, Bingbing Dai, Meng Chen, Fangran Xin, Tingting Wei, Bingqing Bai, Zhijun Fan, Jiahui Li, Yuxin Yao, Ruobing Liao, Jintao Zhang, Xiangnan Jin, and Lingyu Fu. The prognostic value of whole-genome DNA methylation in response to leflunomide in patients with rheumatoid arthritis. *Front. Immunol.*, 14:1173187, September 2023b.

Yung-Che Chen, Chang-Chun Hsiao, Ting-Wen Chen, Chao-Chien Wu, Tung-Ying Chao, Sum-Yee Leung, Hock-Liew Eng, Chiu-Ping Lee, Ting-Ya Wang, and Meng-Chih Lin. Whole genome DNA methylation analysis of active pulmonary tuberculosis disease identifies novel epigenotypes: PARP9/miR-505/RASGRP4/GNG12 gene methylation and clinical phenotypes. *Int. J. Mol. Sci.*, 21(9):3180, April 2020.

Yung-Che Chen, Ying-Huang Tsai, Chin-Chou Wang, Shih-Feng Liu, Ting-Wen Chen, Wen-Feng Fang, Chiu-Ping Lee, Po-Yuan Hsu, Tung-Ying Chao, Chao-Chien Wu, Yu-Feng Wei, Huang-Chih Chang, Chia-Cheng Tsen, Yu-Ping Chang, Meng-Chih Lin, and Taiwan Clinical Trial Consortium of Respiratory Disease (TCORE) group. Epigenome-wide association study on asthma and chronic obstructive pulmonary disease overlap reveals aberrant DNA methylations related to clinical phenotypes. *Sci. Rep.*, 11(1):5022, March 2021.

N Cherny, R Catane, D Schrijvers, M Kloke, and F Strasser. European society for medical oncology (ESMO) program for the integration of oncology and palliative care: a 5-year review of the designated centers' incentive program. *Ann. Oncol.*, 21(2):362–369, February 2010.

Ying-Ming Chiu, Yi-Peng Lu, Joung-Liang Lan, Der-Yuan Chen, and Jung-Der Wang. Lifetime risks, life expectancy, and health care expenditures for rheumatoid arthritis: A nationwide cohort followed up from 2003 to 2016. *Arthritis Rheumatol.*, 73(5):750–758, May 2021.

Michelle Chua, Doyun Kim, Jongmun Choi, Nahyoung G Lee, Vikram Deshpande, Joseph Schwab, Michael H Lev, Ramon G Gonzalez, Michael S Gee, and Synho Do. Tackling prediction uncertainty in machine learning for healthcare. *Nature Biomedical Engineering*, 7(6):711–718, 2023.

Yu-Hsuan Chuang, Kimberly C Paul, Jeff M Bronstein, Yvette Bordelon, Steve Horvath, and Beate Ritz. Parkinson's disease is associated with DNA methylation levels in human blood and saliva. *Genome Med.*, 9 (1):76, August 2017.

Yu-Hsuan Chuang, Ake T Lu, Kimberly C Paul, Aline D Folle, Jeff M Bronstein, Yvette Bordelon, Steve Horvath, and Beate Ritz. Longitudinal epigenome-wide methylation study of cognitive decline and motor progression in parkinson's disease. *J. Parkinsons. Dis.*, 9(2):389–400, 2019.

Alexander D Clark, Nisha Nair, Amy E Anderson, Nishanthi Thalayasingam, Najib Naamane, Andrew J Skelton, Julie Diboll, Anne Barton, Stephen Eyre, John D Isaacs, Arthur G Pratt, and Louise N Reynard. Lymphocyte DNA methylation mediates genetic risk at shared immune-mediated disease loci. *J. Allergy Clin. Immunol.*, 145(5):1438–1451, May 2020.

João Costa, Joana Alarcão, Francisco Araujo, Raquel Ascenção, Daniel Caldeira, Francesca Fiorentino, Victor Gil, Miguel Gouveia, Francisco Lourenço, Alberto Mello E Silva, Filipa Sampaio, António Vaz Carneiro, and Margarida Borges. The burden of atherosclerosis in portugal. *Eur. Heart J. Qual. Care Clin. Outcomes*, 7(2):154–162, March 2021.

Natalia Cullell, Carolina Soriano-Tárraga, Cristina Gallego-Fábrega, Jara Cárcel-Márquez, Elena Muiño, Laia Llucià-Carol, Miquel Lledós, Manel Esteller, Manuel Castro de Moura, Joan Montaner, Anna Rosell, Pilar Delgado, Joan Martí-Fábregas, Jerzy Krupinski, Jaume Roquer, Jordi Jiménez-Conde, and Israel Fernández-Cadenas. Altered methylation pattern in EXOC4 is associated with stroke outcome: an epigenome-wide association study. *Clin. Epigenetics*, 14(1):124, September 2022.

Luke C Dabin, Fernando Guntoro, Tracy Campbell, Tony Bélicard, Adam R Smith, Rebecca G Smith, Rachel Raybould, Jonathan M Schott, Katie Lunnon, Peter Sarkies, John Collinge, Simon Mead, and Emmanuelle Viré. Altered DNA methylation profiles in blood from patients with sporadic Creutzfeldt-Jakob disease. *Acta Neuropathol.*, 140(6):863–879, December 2020.

Haijiang Dai, Arsalan Abu Much, Elad Maor, Elad Asher, Arwa Younis, Yawen Xu, Yao Lu, Xinyao Liu, Jingxian Shu, and Nicola Luigi Bragazzi. Global, regional, and national burden of ischaemic heart disease and its attributable risk factors, 1990-2017: results from the global burden of disease study 2017. *Eur. Heart J. Qual. Care Clin. Outcomes*, 8(1):50–60, January 2022.

Annette Dam Fialla, Ove B Schaffalitzky de Muckadell, and Annmarie Touborg Lassen. Incidence, etiology and mortality of cirrhosis: a population-based cohort study. *Scand. J. Gastroenterol.*, 47(6):702–709, June 2012.

Kenneth Day, Lindsay L Waite, Anna Thalacker-Mercer, Andrew West, Marcas M Bamman, James D Brooks, Richard M Myers, and Devin Absher. Differential DNA methylation with age displays both common and dynamic features across human tissues that are influenced by CpG landscape. *Genome Biol.*, 14(9):R102, 2013.

Carlos de la Calle-Fabregat, Ellis Niemantsverdriet, Juan D Cañete, Tianlu Li, Annette H M van der Helm-van Mil, Javier Rodríguez-Ubreva, and Esteban Ballestar. Prediction of the progression of undifferentiated arthritis to rheumatoid arthritis using DNA methylation profiling. *Arthritis Rheumatol.*, 73(12):2229–2239, December 2021.

Andrew R DiNardo, Kimal Rajapakshe, Tomoki Nishiguchi, Sandra L Grimm, Godwin Mtetwa, Qiniso Dlamini, Jaqueline Kahari, Sanjana Mahapatra, Alexander Kay, Gugu Maphalala, Emily M Mace, George Makedonas, Jeffrey D Cirillo, Mihai G Netea, Reinout van Crevel, Cristian Coarfa, and Anna M Mandalakas. DNA hypermethylation during tuberculosis dampens host immune responsiveness. *J. Clin. Invest.*, 130(6): 3113–3123, June 2020.

Hiroko H Dodge, Changyu Shen, Rajesh Pandav, Steven T DeKosky, and Mary Ganguli. Functional transitions and active life expectancy associated with alzheimer disease. *Arch. Neurol.*, 60(2):253–259, February 2003.

Catherine Dong, Marie Metzger, Einar Holsbø, Vittorio Perduca, and Franck Carbonnel. Systematic review with meta-analysis: mortality in acute severe ulcerative colitis. *Aliment. Pharmacol. Ther.*, 51(1):8–33, January 2020.

Pan Du, Xiao Zhang, Chiang-Ching Huang, Nadereh Jafari, Warren A Kibbe, Lifang Hou, and Simon M Lin. Comparison of beta-value and m-value methods for quantifying methylation levels by microarray analysis. *BMC bioinformatics*, 11:1–9, 2010.

Dana Duricova, Natalia Pedersen, Margarita Elkjaer, Michael Gamborg, Pia Munkholm, and Tine Jess. Overall and cause-specific mortality in crohn's disease: a meta-analysis of population-based studies. *Inflamm. Bowel Dis.*, 16(2):347–353, February 2010.

Emerging Risk Factors Collaboration. Life expectancy associated with different ages at diagnosis of type 2 diabetes in high-income countries: 23 million person-years of observation. *Lancet Diabetes Endocrinol.*, 11 (10):731–742, October 2023.

Andrés Esteban-Cantos, Javier Rodríguez-Centeno, Juan C Silla, Pilar Barruz, Fátima Sánchez-Cabo, Gabriel Saiz-Medrano, Julián Nevado, Beatriz Mena-Garay, María Jiménez-González, Rosa de Miguel, Jose I Bernardino, Rocío Montejano, Julen Cadiñanos, Cristina Marcelo, Lucía Gutiérrez-García, Patricia Martínez-Martín, Cédrick Wallet, François Raffi, Berta Rodés, José R Arribas, and NEAT001/ANRS143 study group. Effect of HIV infection and antiretroviral therapy initiation on genome-wide DNA methylation patterns. *EBioMedicine*, 88(104434):104434, February 2023.

Maria M Esterhuyse, January Weiner, 3rd, Etienne Caron, Andre G Loxton, Marco Iannaccone, Chandre Wagman, Philippe Saikali, Kim Stanley, Witold E Wolski, Hans-Joachim Mollenkopf, Matthias Schick, Ruedi Aebersold, Heinz Linhart, Gerhard Walzl, and Stefan H E Kaufmann. Epigenetics and proteomics join transcriptomics in the quest for tuberculosis biomarkers. *MBio*, 6(5):e01187–15, September 2015.

Ewoud Ewing, Lara Kular, Sunjay J Fernandes, Nestoras Karathanasis, Vincenzo Lagani, Sabrina Ruhrmann, Ioannis Tsamardinos, Jesper Tegner, Fredrik Piehl, David Gomez-Cabrero, and Maja Jagodic. Combining evidence from four immune cell types identifies DNA methylation patterns that implicate functionally distinct pathways during multiple sclerosis progression. *EBioMedicine*, 43:411–423, May 2019.

Evans R Fernández Pérez, Craig E Daniels, Darrell R Schroeder, Jennifer St Sauver, Thomas E Hartman, Brian J Bartholmai, Eunhee S Yi, and Jay H Ryu. Incidence, prevalence, and clinical course of idiopathic pulmonary fibrosis: a population-based study. *Chest*, 137(1):129–137, January 2010.

Eduardo Fernandez-Rebollo, Monika Eipel, Lothar Seefried, Per Hoffmann, Klaus Strathmann, Franz Jakob, and Wolfgang Wagner. Primary osteoporosis is not reflected by disease-specific DNA methylation or accelerated epigenetic age in blood. *J. Bone Miner. Res.*, 33(2):356–361, February 2018.

Kevin R Fontaine, David T Redden, Chenxi Wang, Andrew O Westfall, and David B Allison. Years of life lost due to obesity. *JAMA*, 289(2):187–193, January 2003.

Ming Fu, Hongming Zhou, Yushi Li, Hai Jin, and Xiqing Liu. Global, regional, and national burdens of hip osteoarthritis from 1990 to 2019: estimates from the 2019 global burden of disease study. *Arthritis Res. Ther.*, 24(1):8, January 2022.

Mary Ganguli, Hiroko H Dodge, Changyu Shen, Rajesh S Pandav, and Steven T DeKosky. Alzheimer disease and mortality: a 15-year epidemiological study. *Arch. Neurol.*, 62(5):779–784, May 2005.

GBD 2019 Stroke Collaborators. Global, regional, and national burden of stroke and its risk factors, 1990-2019: a systematic analysis for the global burden of disease study 2019. *Lancet Neurol.*, 20(10):795–820, October 2021.

Ellen Gelpi, Harald Heinzl, Romana Hoftberger, Ursula Unterberger, Thomas Strobel, Till Voigtlander, Edita Drobna, Christa Jarius, Susanna Lang, Thomas Waldhor, Hanno Bernheimer, and Herbert Budka. Creutzfeldt-Jakob disease in austria: an autopsy-controlled study. *Neuroepidemiology*, 30(4):215–221, April 2008.

Rodney C P Go, Michael J Corley, G Webster Ross, Helen Petrovitch, Kamal H Masaki, Alika K Maunakea, Qimei He, and Maarit I Tiirikainen. Genome-wide epigenetic analyses in japanese immigrant plantation workers with parkinson's disease and exposure to organochlorines reveal possible involvement of glial genes and pathways involved in neurotoxicity. *BMC Neurosci.*, 21(1):31, July 2020.

M Goto. Hierarchical deterioration of body systems in werner's syndrome: implications for normal ageing. *Mech. Ageing Dev.*, 98(3):239–254, December 1997.

Andrew M Gross, Philipp A Jaeger, Jason F Kreisberg, Katherine Licon, Kristen L Jepsen, Mahdieh Khosro-heidari, Brenda M Morsey, Susan Swindells, Hui Shen, Cherie T Ng, Ken Flagg, Daniel Chen, Kang Zhang, Howard S Fox, and Trey Ideker. Methylome-wide analysis of chronic HIV infection reveals five-year increase in biological age and epigenetic targeting of HLA. *Mol. Cell*, 62(2):157–168, April 2016.

Rebbeca A Grysiewicz, Kurian Thomas, and Dilip K Pandey. Epidemiology of ischemic and hemorrhagic stroke: incidence, prevalence, mortality, and risk factors. *Neurol. Clin.*, 26(4):871–95, vii, November 2008.

S Gyde, P Prior, M J Dew, V Saunders, J A Waterhouse, and R N Allan. Mortality in ulcerative colitis. *Gastroenterology*, 83(1 Pt 1):36–43, July 1982.

Gregory Hannum, Justin Guinney, Ling Zhao, LI Zhang, Guy Hughes, SriniVas Sadda, Brandy Klotzle, Marina Bibikova, Jian-Bing Fan, Yuan Gao, et al. Genome-wide methylation profiles reveal quantitative views of human aging rates. *Molecular cell*, 49(2):359–367, 2013.

Valma Harjutsalo, Carol Forsblom, and Per-Henrik Groop. Time trends in mortality in patients with type 1 diabetes: nationwide population based cohort study. *BMJ*, 343(sep08 2):d5364, September 2011.

R Alan Harris, Dorottya Nagy-Szakal, Natalia Pedersen, Antone Opekun, Jiri Bronsky, Pia Munkholm, Cathrine Jespersgaard, Paalskytt Andersen, Bela Melegh, George Ferry, Tine Jess, and Richard Kellermayer. Genome-wide peripheral blood leukocyte DNA methylation microarrays identified a single association with inflammatory bowel diseases. *Inflamm. Bowel Dis.*, 18(12):2334–2341, December 2012.

Adam Hartley, Dominic C Marshall, Justin D Salciccioli, Markus B Sikkel, Mahiben Maruthappu, and Joseph Shalhoub. Trends in mortality from ischemic heart disease and cerebrovascular disease in europe: 1980 to 2009. *Circulation*, 133(20):1916–1926, May 2016.

Adrian H Heald, Mike Stedman, Mark Davies, Mark Livingston, Ramadan Alshames, Mark Lunt, Gerry Rayman, and Roger Gadsby. Estimating life years lost to diabetes: outcomes from analysis of national diabetes audit and office of national statistics data. *Cardiovasc. Endocrinol. Metab.*, 9(4):183–185, December 2020.

Raoul C M Hennekam. Hutchinson-Gilford progeria syndrome: review of the phenotype. *Am. J. Med. Genet. A*, 140(23):2603–2624, December 2006.

Johannes Hertel, Nele Friedrich, Katharina Wittfeld, Maik Pietzner, Kathrin Budde, Sandra Van der Auwera, Tobias Lohmann, Alexander Teumer, Henry Voolzke, Matthias Nauck, et al. Measuring biological age via metabonomics: the metabolic age score. *Journal of proteome research*, 15(2):400–410, 2016.

Albert T Higgins-Chen, Kyra L Thrush, Yunzhang Wang, Christopher J Minteer, Pei-Lun Kuo, Meng Wang, Peter Niimi, Gabriel Sturm, Jue Lin, Ann Zenobia Moore, et al. A computational solution for bolstering reliability of epigenetic clocks: Implications for clinical trials and longitudinal tracking. *Nature aging*, 2(7): 644–661, 2022.

Nicholas Holzscheck, Cassandra Falckenhayn, Jörn Söhle, Boris Kristof, Ralf Siegner, André Werner, Janka Schössow, Clemens Jürgens, Henry Völzke, Horst Wenck, et al. Modeling transcriptomic age using knowledge-primed artificial neural networks. *npj Aging and Mechanisms of Disease*, 7(1):15, 2021.

Steve Horvath. Dna methylation age of human tissues and cell types. *Genome biology*, 14:1–20, 2013.

Steve Horvath and Andrew J Levine. HIV-1 infection accelerates age according to the epigenetic clock. *J. Infect. Dis.*, 212(10):1563–1573, November 2015.

Steve Horvath and Beate R Ritz. Increased epigenetic age and granulocyte counts in the blood of parkinson's disease patients. *Aging (Albany NY)*, 7(12):1130–1142, December 2015.

Steve Horvath, Junko Oshima, George M Martin, Ake T Lu, Austin Quach, Howard Cohen, Sarah Felton, Mieko Matsuyama, Donna Lowe, Sylwia Kabacik, et al. Epigenetic clock for skin and blood cells applied to hutchinson gilford progeria syndrome and ex vivo studies. *Aging (Albany NY)*, 10(7):1758, 2018.

Tianxiao Huan, Steve Nguyen, Elena Colicino, Carolina Ochoa-Rosales, W David Hill, Jennifer A Brody, Mette Soerensen, Yan Zhang, Antoine Baldassari, Mohamed Ahmed Elhadad, et al. Integrative analysis of clinical and epigenetic biomarkers of mortality. *Aging cell*, 21(6):e13608, 2022.

Lili Huo, Jessica L Harding, Anna Peeters, Jonathan E Shaw, and Dianna J Magliano. Life expectancy of type 1 diabetic patients during 1997-2010: a national australian registry-based cohort study. *Diabetologia*, 59(6): 1177–1185, June 2016.

John P Hutchinson, Tricia M McKeever, Andrew W Fogarty, Vidya Navaratnam, and Richard B Hubbard. Increasing global mortality from idiopathic pulmonary fibrosis in the twenty-first century. *Ann. Am. Thorac. Soc.*, 11(8):1176–1185, October 2014.

Yoshiyuki Ikeda and Mitsuru Ohishi. Years of life lost analysis may promote governmental policy to prevent atherosclerotic cardiovascular disease. *Circ. J.*, 83(5):965–966, April 2019.

Geoffrey Istas, Ken Declerck, Maria Pudenz, Katarzyna Szarc Vel Szic, Veronica Lendinez-Tortajada, Montserrat Leon-Latre, Karen Heyninck, Guy Haegeman, Jose A Casasnovas, Maria Tellez-Plaza, Clarissa Gerhauser, Christian Heiss, Ana Rodriguez-Mateos, and Wim Vanden Berghe. Identification of differentially methylated BRCA1 and CRISP2 DNA regions as blood surrogate markers for cardiovascular disease. *Sci. Rep.*, 7(1):5120, July 2017.

L T Jacobsson, W C Knowler, S Pillemer, R L Hanson, D J Pettitt, R G Nelson, A del Puente, D R McCance, M A Charles, and P H Bennett. Rheumatoid arthritis and mortality. a longitudinal study in pima indians. *Arthritis Rheum.*, 36(8):1045–1053, August 1993.

Antonio Julià, Antonio Gómez, María López-Lasanta, Francisco Blanco, Alba Erra, Antonio Fernández-Nebro, Antonio Juan Mas, Carolina Pérez-García, Ma Luz García Vivar, Simón Sánchez-Fernández, Mercedes Alperi-López, Raimon Sanmartí, Ana María Ortiz, Carlos Marras Fernandez-Cid, César Díaz-Torné, Estefania Moreno, Tianlu Li, Sergio H Martínez-Mateu, Devin M Absher, Richard M Myers, Jesús Tornero Molina, and Sara Marsal. Longitudinal analysis of blood DNA methylation identifies mechanisms of response to tumor necrosis factor inhibitor therapy in rheumatoid arthritis. *EBioMedicine*, 80(104053):104053, June 2022.

Aki Juhani Käräjämäki, Arto Korkiakoski, Janne Hukkanen, Y Antero Kesäniemi, and Olavi Ukkola. Long-term metabolic fate and mortality in obesity without metabolic syndrome. *Ann. Med.*, 54(1):1432–1443, December 2022.

Changrong Ke, Juanjuan Liang, Mi Liu, Shiwei Liu, and Chunping Wang. Burden of chronic kidney disease and its risk-attributable burden in 137 low-and middle-income countries, 1990-2019: results from the global burden of disease study 2019. *BMC Nephrol.*, 23(1):17, January 2022.

Kyeong Min Kim, Hyung Jung Oh, Hyung Yun Choi, Hajeong Lee, and Dong-Ryeol Ryu. Impact of chronic kidney disease on mortality: A nationwide cohort study. *Kidney Res. Clin. Pract.*, 38(3):382–390, September 2019.

Cari M Kitahara, Alan J Flint, Amy Berrington de Gonzalez, Leslie Bernstein, Michelle Brotzman, Robert J MacInnis, Steven C Moore, Kim Robien, Philip S Rosenberg, Pramil N Singh, Elisabete Weiderpass, Hans Olov Adami, Hoda Anton-Culver, Rachel Ballard-Barbash, Julie E Buring, D Michal Freedman, Gary E Fraser, Laura E Beane Freeman, Susan M Gapstur, John Michael Gaziano, Graham G Giles, Niclas Håkansson, Jane A Hoppin, Frank B Hu, Karen Koenig, Martha S Linet, Yikyung Park, Alpa V Patel, Mark P Purdue, Catherine Schairer, Howard D Sesso, Kala Visvanathan, Emily White, Alicja Wolk, Anne Zeleniuch-Jacquotte, and Patricia Hartge. Association between class III obesity (BMI of 40-59 kg/m2) and mortality: a pooled analysis of 20 prospective studies. *PLoS Med.*, 11(7):e1001673, July 2014.

Martin Kolb and Harold R Collard. Staging of idiopathic pulmonary fibrosis: past, present and future. *Eur. Respir. Rev.*, 23(132):220–224, June 2014.

D Kornfeld, A Ekbom, and T Ihre. Survival and risk of cholangiocarcinoma in patients with primary sclerosing cholangitis. a population-based study. *Scand. J. Gastroenterol.*, 32(10):1042–1045, October 1997.

Dmitrii Kriukov, Ekaterina Kuzmina, Evgeniy Efimov, Dmitry V Dylov, and Ekaterina E Khrameeva. Epistemic uncertainty challenges aging clock reliability in predicting rejuvenation effects. *bioRxiv*, pp. 2023–12, 2023.

M Ellen Kuenzig, Douglas G Manuel, Jessy Donelle, and Eric I Benchimol. Life expectancy and health-adjusted life expectancy in people with inflammatory bowel disease. *CMAJ*, 192(45):E1394–E1402, November 2020.

Lara Kular, Yun Liu, Sabrina Ruhrmann, Galina Zheleznyakova, Francesco Marabita, David Gomez-Cabrero, Tojo James, Ewoud Ewing, Magdalena Lindén, Bartosz Górnikiewicz, Shahin Aeinehband, Pernilla Stridh, Jenny Link, Till F M Andlauer, Christiane Gasperi, Heinz Wiendl, Frauke Zipp, Ralf Gold, Björn Tackenberg, Frank Weber, Bernhard Hemmer, Konstantin Strauch, Stefanie Heilmann-Heimbach, Rajesh Rawal, Ulf Schminke, Carsten O Schmidt, Tim Kacprowski, Andre Franke, Matthias Laudes, Alexander T Dilthey, Elisabeth G Celius, Helle B Søndergaard, Jesper Tegnér, Hanne F Harbo, Annette B Oturai, Sigurgeir Olafsson, Hannes P Eggertsson, Bjarni V Halldorsson, Haukur Hjaltason, Elias Olafsson, Ingileif Jonsdottir, Kari Stefansson, Tomas Olsson, Fredrik Piehl, Tomas J Ekström, Ingrid Kockum, Andrew P Feinberg, and Maja Jagodic. DNA methylation as a mediator of HLA-DRB1*15:01 and a protective variant in multiple sclerosis. *Nat. Commun.*, 9(1):2397, June 2018.

Lisa Lancaster, Francesco Bonella, Yoshikazu Inoue, Vincent Cottin, James Siddall, Mark Small, and Jonathan Langley. Idiopathic pulmonary fibrosis: Physician and patient perspectives on the pathway to care from symptom recognition to diagnosis and disease burden. *Respirology*, 27(1):66–75, January 2022.

Scott D Landes, J Dalton Stevens, and Margaret A Turk. Cause of death in adults with down syndrome in the united states. *Disability and health journal*, 13(4):100947, 2020.

Peter Lange, Yunus Çolak, Truls Sylvan Ingebrigtsen, Jørgen Vestbo, and Jacob Louis Marott. Long-term prognosis of asthma, chronic obstructive pulmonary disease, and asthma-chronic obstructive pulmonary disease overlap in the copenhagen city heart study: a prospective population-based analysis. *Lancet Respir. Med.*, 4(6):454–462, June 2016.

M N Lassere, J Rappo, I J Portek, A Sturgess, and J P Edmonds. How many life years are lost in patients with rheumatoid arthritis? secular cause-specific and all-cause mortality in rheumatoid arthritis, and their predictors in a long-term australian cohort study. *Intern. Med. J.*, 43(1):66–72, January 2013.

Christian Lee-Rodriguez, Paul Y Wada, Yun-Yi Hung, and Jacek Skarbinski. Association of mortality and years of potential life lost with active tuberculosis in the united states. *JAMA Netw. Open*, 3(9):e2014481, September 2020.

Rebecca A Legarth, Magnus G Ahlström, Gitte Kronborg, Carsten S Larsen, Court Pedersen, Gitte Pedersen, Rajesh Mohey, Jan Gerstoft, and Niels Obel. Long-term mortality in HIV-infected individuals 50 years or older: A nationwide, population-based cohort study. *J. Acquir. Immune Defic. Syndr.*, 71(2):213–218, February 2016.

Emmanuelle Leray, Sandra Vukusic, Marc Debouverie, Michel Clanet, Bruno Brochet, Jérôme de Sèze, Hélène Zéphir, Gilles Defer, Christine Lebrun-Frenay, Thibault Moreau, Pierre Clavelou, Jean Pelletier, Eric Berger, Philippe Cabre, Jean-Philippe Camdessanché, Shoshannah Kalson-Ray, Christian Confavreux, and Gilles Edan. Excess mortality in patients with multiple sclerosis starts at 20 years from clinical onset: Data from a large-scale french observational study. *PLoS One*, 10(7):e0132033, July 2015.

B Lernfelt, M Forsberg, C Blomstrand, D Mellström, and R Volkmann. Cerebral atherosclerosis as predictor of stroke and mortality in representative elderly population. *Stroke*, 33(1):224–229, January 2002.

Morgan E Levine, Ake T Lu, Austin Quach, Brian H Chen, Themistocles L Assimes, Stefania Bandinelli, Lifang Hou, Andrea A Baccarelli, James D Stewart, Yun Li, et al. An epigenetic biomarker of aging for lifespan and healthspan. *Aging (albany NY)*, 10(4):573, 2018.

Xue Li, Xiaojin Feng, Xiaodong Sun, Ningning Hou, Fang Han, and Yongping Liu. Global, regional, and national burden of alzheimer's disease and other dementias, 1990-2019. *Front. Aging Neurosci.*, 14:937486, October 2022.

Andrew Y F Li Yim, Nicolette W Duijvis, Jing Zhao, Wouter J de Jonge, Geert R A M D'Haens, Marcel M A M Mannens, Adri N P M Mul, Anje A Te Velde, and Peter Henneman. Peripheral blood methylation profiling of female crohn's disease patients. *Clin. Epigenetics*, 8(1):65, June 2016.

Chih-Sung Liang, Dian-Jeng Li, Fu-Chi Yang, Ping-Tao Tseng, Andre F Carvalho, Brendon Stubbs, Trevor Thompson, Christoph Mueller, Jae Il Shin, Joaquim Radua, Robert Stewart, Tarek K Rajji, Yu-Kang Tu, Tien-Yu Chen, Ta-Chuan Yeh, Chia-Kuang Tsai, Chia-Ling Yu, Chih-Chuan Pan, and Che-Sheng Chu. Mortality rates in alzheimer's disease and non-alzheimer's dementias: a systematic review and meta-analysis. *Lancet Healthy Longev.*, 2(8):e479–e488, August 2021a.

Chih-Sung Liang, Dian-Jeng Li, Fu-Chi Yang, Ping-Tao Tseng, Andre F Carvalho, Brendon Stubbs, Trevor Thompson, Christoph Mueller, Jae Il Shin, Joaquim Radua, Robert Stewart, Tarek K Rajji, Yu-Kang Tu, Tien-Yu Chen, Ta-Chuan Yeh, Chia-Kuang Tsai, Chia-Ling Yu, Chih-Chuan Pan, and Che-Sheng Chu. Mortality rates in alzheimer's disease and non-alzheimer's dementias: a systematic review and meta-analysis. *Lancet Healthy Longev.*, 2(8):e479–e488, August 2021b.

Josivan Gomes Lima, Lucia Helena C Nobrega, Natalia Nobrega Lima, Marcel Catão Ferreira Dos Santos, Pedro Henrique Dantas Silva, Maria de Fatima P Baracho, Debora Nobrega Lima, Julliane Tamara Araújo de Melo Campos, Leonardo Capistrano Ferreira, Francisco Paulo Freire Neto, Carolina de O Mendes-Aguiar, and Selma Maria B Jeronimo. Causes of death in patients with Berardinelli-Seip congenital generalized lipodystrophy. *PLoS One*, 13(6):e0199052, June 2018.

Qiong Lin and Wolfgang Wagner. Epigenetic aging signatures are coherently modified in cancer. *PLoS genetics*, 11(6):e1005334, 2015.

Qiong Lin, Carola I Weidner, Ivan G Costa, Riccardo E Marioni, Marcelo RP Ferreira, Ian J Deary, and Wolfgang Wagner. Dna methylation levels at individual age-associated cpg sites can be indicative for life expectancy. *Aging (Albany NY)*, 8(2):394, 2016.

Minbo Liu, Fang Jin, Xiaocong Yao, and Zhongxin Zhu. Disease burden of osteoarthritis of the knee and hip due to a high body mass index in china and the USA: 1990-2019 findings from the global burden of disease study 2019. *BMC Musculoskelet. Disord.*, 23(1):63, January 2022.

Q Liu, J Niu, J Huang, Y Ke, X Tang, X Wu, R Li, H Li, X Zhi, K Wang, Y Zhang, and J Lin. Knee osteoarthritis and all-cause mortality: the wuchuan osteoarthritis study. *Osteoarthritis Cartilage*, 23(7):1154–1157, July 2015.

Yun Liu, Martin J Aryee, Leonid Padyukov, M Daniele Fallin, Espen Hesselberg, Arni Runarsson, Lovisa Reinius, Nathalie Acevedo, Margaret Taub, Marcus Ronninger, Klementy Shchetynsky, Annika Scheynius, Juha Kere, Lars Alfredsson, Lars Klareskog, Tomas J Ekström, and Andrew P Feinberg. Epigenome-wide association data implicate DNA methylation as an intermediary of genetic risk in rheumatoid arthritis. *Nat. Biotechnol.*, 31(2):142–147, February 2013.

Shona J Livingstone, Daniel Levin, Helen C Looker, Robert S Lindsay, Sarah H Wild, Nicola Joss, Graham Leese, Peter Leslie, Rory J McCrimmon, Wendy Metcalfe, John A McKnight, Andrew D Morris, Donald W M Pearson, John R Petrie, Sam Philip, Naveed A Sattar, Jamie P Traynor, Helen M Colhoun, Scottish Diabetes Research Network epidemiology group, and Scottish Renal Registry. Estimated life expectancy in a scottish cohort with type 1 diabetes, 2008-2010. *JAMA*, 313(1):37–44, January 2015.

Ana Lleo, Peter Jepsen, Emanuela Morenghi, Marco Carbone, Luca Moroni, Pier Maria Battezzati, Mauro Podda, Ian R Mackay, M Eric Gershwin, and Pietro Invernizzi. Evolving trends in female to male incidence and male mortality of primary biliary cholangitis. *Sci. Rep.*, 6:25906, May 2016.

Franc Llorens, Nicole Rübsamen, Peter Hermann, Matthias Schmitz, Anna Villar-Piqué, Stefan Goebel, André Karch, and Inga Zerr. A prognostic model for overall survival in sporadic Creutzfeldt-Jakob disease. *Alzheimers. Dement.*, 16(10):1438–1447, October 2020.

Ake T Lu, Austin Quach, James G Wilson, Alex P Reiner, Abraham Aviv, Kenneth Raj, Lifang Hou, Andrea A Baccarelli, Yun Li, James D Stewart, et al. Dna methylation grimage strongly predicts lifespan and healthspan. *Aging (albany NY)*, 11(2):303, 2019.

Ake T Lu, Alexandra M Binder, Joshua Zhang, Qi Yan, Alex P Reiner, Simon R Cox, Janie Corley, Sarah E Harris, Pei-Lun Kuo, Ann Z Moore, et al. Dna methylation grimage version 2. *Aging (Albany NY)*, 14(23): 9484, 2022.

Hanne Marie Bøe Lunde, Jörg Assmus, Kjell-Morten Myhr, Lars Bø, and Nina Grytten. Survival and cause of death in multiple sclerosis: a 60-year longitudinal population study. *J. Neurol. Neurosurg. Psychiatry*, 88 (8):621–625, August 2017.

Katie Lunnon, Rebecca Smith, Eilis Hannon, Philip L De Jager, Gyan Srivastava, Manuela Volta, Claire Troakes, Safa Al-Sarraj, Joe Burrage, Ruby Macdonald, Daniel Condliffe, Lorna W Harries, Pavel Katsel, Vahram Haroutunian, Zachary Kaminsky, Catharine Joachim, John Powell, Simon Lovestone, David A Bennett, Leonard C Schalkwyk, and Jonathan Mill. Methylomic profiling implicates cortical deregulation of ANK1 in alzheimer's disease. *Nat. Neurosci.*, 17(9):1164–1170, September 2014.

Katie Lunnon, Rebecca G Smith, Itzik Cooper, Lior Greenbaum, Jonathan Mill, and Michal Schnaider Beeri. Blood methylomic signatures of presymptomatic dementia in elderly subjects with type 2 diabetes mellitus. *Neurobiol. Aging*, 36(3):1600.e1–4, March 2015.

Angus D Macleod, Kate S M Taylor, and Carl E Counsell. Mortality in parkinson's disease: a systematic review and meta-analysis. *Mov. Disord.*, 29(13):1615–1622, November 2014.

Anna Maierhofer, Julia Flunkert, Junko Oshima, George M Martin, Martin Poot, Indrajit Nanda, Marcus Dittrich, Tobias Müller, and Thomas Haaf. Epigenetic signatures of werner syndrome occur early in life and are distinct from normal epigenetic aging processes. *Aging Cell*, 18(5):e12995, October 2019.

Francesco Marabita, Malin Almgren, Maléne E Lindholm, Sabrina Ruhrmann, Fredrik Fagerström-Billai, Maja Jagodic, Carl J Sundberg, Tomas J Ekström, Andrew E Teschendorff, Jesper Tegnér, and David Gomez-Cabrero. An evaluation of analysis pipelines for DNA methylation profiling using the illumina Human-Methylation450 BeadChip platform. *Epigenetics*, 8(3):333–346, March 2013.

Seth S Martin, Aaron W Aday, Zaid I Almarzooq, Cheryl A M Anderson, Pankaj Arora, Christy L Avery, Carissa M Baker-Smith, Bethany Barone Gibbs, Andrea Z Beaton, Amelia K Boehme, Yvonne Commodore-Mensah, Maria E Currie, Mitchell S V Elkind, Kelly R Evenson, Giuliano Generoso, Debra G Heard, Swapnil Hiremath, Michelle C Johansen, Rizwan Kalani, Dhruv S Kazi, Darae Ko, Junxiu Liu, Jared W Magnani, Erin D Michos, Michael E Mussolino, Sankar D Navaneethan, Nisha I Parikh, Sarah M Perman, Remy Poudel, Mary Rezk-Hanna, Gregory A Roth, Nilay S Shah, Marie-Pierre St-Onge, Evan L Thacker, Connie W Tsao, Sarah M Urbut, Harriette G C Van Spall, Jenifer H Voeks, Nae-Yuh Wang, Nathan D Wong, Sally S Wong, Kristine Yaffe, Latha P Palaniappan, and American Heart Association Council on Epidemiology and Prevention Statistics Committee and Stroke Statistics Subcommittee. 2024 heart disease and stroke statistics: A report of US and global data from the american heart association. *Circulation*, 149(8): e347–e913, February 2024.

Ryan K Masters, Eric N Reither, Daniel A Powers, Y Claire Yang, Andrew E Burger, and Bruce G Link. The impact of obesity on US mortality levels: the importance of age and cohort factors in population estimates. *Am. J. Public Health*, 103(10):1895–1901, October 2013.

Margaret T May, Mark Gompels, Valerie Delpech, Kholoud Porter, Chloe Orkin, Stephen Kegg, Phillip Hay, Margaret Johnson, Adrian Palfreeman, Richard Gilson, David Chadwick, Fabiola Martin, Teresa Hill, John Walsh, Frank Post, Martin Fisher, Jonathan Ainsworth, Sophie Jose, Clifford Leen, Mark Nelson, Jane Anderson, Caroline Sabin, and UK Collaborative HIV Cohort (UK CHIC) Study. Impact on life expectancy of HIV-1 positive individuals of CD4+ cell count and viral load response to antiretroviral therapy. *AIDS*, 28 (8):1193–1202, May 2014.

Xiaoyue Mei, Joshua Blanchard, Connor Luellen, Michael J Conboy, and Irina M Conboy. Fail-tests of dna methylation clocks, and development of a noise barometer for measuring epigenetic pressure of aging and disease. *Aging (Albany NY)*, 15(17):8552, 2023.

Nicolas A Menzies, Matthew Quaife, Brian W Allwood, Anthony L Byrne, Anna K Coussens, Anthony D Harries, Florian M Marx, Jamilah Meghji, Debora Pedrazzoli, Joshua A Salomon, Sedona Sweeney, Sanne C van Kampen, Robert S Wallis, Rein M G J Houben, and Ted Cohen. Lifetime burden of disease due to incident tuberculosis: a global reappraisal including post-tuberculosis sequelae. *Lancet Glob. Health*, 9(12): e1679–e1687, December 2021.

Mahdi Moqri, Chiara Herzog, Jesse R Poganik, Kejun Ying, Jamie N Justice, Daniel W Belsky, Albert T Higgins-Chen, Brian H Chen, Alan A Cohen, Georg Fuellen, et al. Validation of biomarkers of aging. *Nature Medicine*, pp. 1–13, 2024.

Francesco Morandini, Cheyenne Rechsteiner, Kevin Perez, Viviane Praz, Guillermo Lopez Garcia, Laura C Hinte, Ferdinand von Meyenn, and Alejandro Ocampo. Atac-clock: An aging clock based on chromatin accessibility. *GeroScience*, 46(2):1789–1806, 2024.

Dariush Mozaffarian, Aruna Kamineni, Ronald J Prineas, and David S Siscovick. Metabolic syndrome and mortality in older adults: the cardiovascular health study. *Arch. Intern. Med.*, 168(9):969–978, May 2008.

Christoph Mueller, Pinar Soysal, Arvid Rongve, Ahmet Turan Isik, Trevor Thompson, Stefania Maggi, Lee Smith, Cristina Basso, Robert Stewart, Clive Ballard, John T O'Brien, Dag Aarsland, Brendon Stubbs, and Nicola Veronese. Survival time and differences between dementia with lewy bodies and alzheimer's disease following diagnosis: A meta-analysis of longitudinal studies. *Ageing Res. Rev.*, 50:72–80, March 2019.

H E Mulnier, H E Seaman, V S Raleigh, S S Soedamah-Muthu, H M Colhoun, and R A Lawrenson. Mortality in people with type 2 diabetes in the UK. *Diabet. Med.*, 23(5):516–521, May 2006.

Fumiyo Nakagawa, Margaret May, and Andrew Phillips. Life expectancy living with HIV: recent estimates and future implications. *Curr. Opin. Infect. Dis.*, 26(1):17–25, February 2013.

Maria S Nazarenko, Anton V Markov, Igor N Lebedev, Maxim B Freidin, Aleksei A Sleptcov, Iuliya A Koroleva, Aleksei V Frolov, Vadim A Popov, Olga L Barbarash, and Valery P Puzyrev. A comparison of genomewide DNA methylation patterns between different vascular tissues from patients with coronary heart disease. *PLoS One*, 10(4):e0122601, April 2015.

Yoshito Nishimura, Ko Harada, Toshihiro Koyama, Hideharu Hagiya, and Fumio Otsuka. A nationwide trend analysis in the incidence and mortality of Creutzfeldt-Jakob disease in japan between 2005 and 2014. *Sci. Rep.*, 10(1):15509, September 2020.

Natália Yumi Noronha, Mariana Barato, Chanachai Sae-Lee, Marcela Augusta de Souza Pinhel, Lígia Moriguchi Watanabe, Vanessa Aparecida Batista Pereira, Guilherme da Silva Rodrigues, Déborah Araújo Morais, Wellington Tavares de Sousa, Jr, Vanessa Cristina de Oliveira Souza, Jessica Rodrigues Plaça, Wilson Salgado, Jr, Fernando Barbosa, Jr, Torsten Plösch, and Carla Barbosa Nonino. Novel zinc-related differentially methylated regions in leukocytes of women with and without obesity. *Front. Nutr.*, 9:785281, March 2022.

Achilles Ntranos, Vasilis Ntranos, Valentina Bonnefil, Jia Liu, Seunghee Kim-Schulze, Ye He, Yunjiao Zhu, Rachel Brandstadter, Corey T Watson, Andrew J Sharp, Ilana Katz Sand, and Patrizia Casaccia. Fumarates target the metabolic-epigenetic interplay of brain-homing T cells in multiple sclerosis. *Brain*, 142(3):647–661, March 2019.

Padraig Oakley, Steve Kisely, Amanda Baxter, Meredith Harris, Jocelyne Desoe, Alyona Dziouba, and Dan Siskind. Increased mortality among people with schizophrenia and other non-affective psychotic disorders in the community: a systematic review and meta-analysis. *Journal of psychiatric research*, 102:245–253, 2018.

Lisa O'Leary, Laura Hughes-McCormack, Kirsty Dunn, and Sally-Ann Cooper. Early death and causes of death of people with down syndrome: a systematic review. *Journal of Applied Research in Intellectual Disabilities*, 31(5):687–708, 2018.

Mark Olfson, Tobias Gerhard, Cecilia Huang, Stephen Crystal, and T Scott Stroup. Premature mortality among adults with schizophrenia in the united states. *JAMA psychiatry*, 72(12):1172–1181, 2015.

Bruna Oriol-Tordera, Maria Berdasco, Anuska Llano, Beatriz Mothe, Cristina Gálvez, Javier Martinez-Picado, Jorge Carrillo, Julià Blanco, Clara Duran-Castells, Carmela Ganoza, Jorge Sanchez, Bonaventura Clotet, Maria Luz Calle, Alex Sánchez-Pla, Manel Esteller, Christian Brander, and Marta Ruiz-Riol. Methylation regulation of antiviral host factors, interferon stimulated genes (ISGs) and t-cell responses associated with natural HIV control. *PLoS Pathog.*, 16(8):e1008678, August 2020.

Bruna Oriol-Tordera, Anna Esteve-Codina, María Berdasco, Míriam Rosás-Umbert, Elena Gonçalves, Clara Duran-Castells, Francesc Català-Moll, Anuska Llano, Samandhy Cedeño, Maria C Puertas, Martin Tolstrup, Ole S Søgaard, Bonaventura Clotet, Javier Martínez-Picado, Tomáš Hanke, Behazine Combadiere, Roger Paredes, Dennis Hartigan-O'Connor, Manel Esteller, Michael Meulbroek, María Luz Calle, Alex Sanchez-Pla, José Moltó, Beatriz Mothe, Christian Brander, and Marta Ruiz-Riol. Epigenetic landscape in the kick-and-kill therapeutic vaccine BCN02 clinical trial is associated with antiretroviral treatment interruption (ATI) outcome. *EBioMedicine*, 78(103956):103956, April 2022.

Junko Oshima and Fuki M Hisama. Search and insights into novel genetic alterations leading to classical and atypical werner syndrome. *Gerontology*, 60(3):239–246, January 2014.

Seon Cheol Park, Dong Wook Kim, Eun Cheol Park, Cheung Soo Shin, Chin Kook Rhee, Young Ae Kang, and Young Sam Kim. Mortality of patients with chronic obstructive pulmonary disease: a nationwide populationbased cohort study. *Korean J. Intern. Med.*, 34(6):1272–1278, November 2019.

Ignacio J Posada, Julián Benito-León, Elan D Louis, Rocío Trincado, Alberto Villarejo, María José Medrano, and Félix Bermejo-Pareja. Mortality from parkinson's disease: a population-based prospective study (NEDICES). *Mov. Disord.*, 26(14):2522–2529, December 2011.

Angela P Presson, Ginger Partyka, Kristin M Jensen, Owen J Devine, Sonja A Rasmussen, Linda L McCabe, and Edward RB McCabe. Current estimate of down syndrome population prevalence in the united states. *The Journal of pediatrics*, 163(4):1163–1168, 2013.

Annabel Price, Redwan Farooq, Jin-Min Yuan, Vandana B Menon, Rudolf N Cardinal, and John T O'Brien. Mortality in dementia with lewy bodies compared with alzheimer's dementia: a retrospective naturalistic cohort study. *BMJ Open*, 7(11):e017504, November 2017.

Jonathan Q Purnell. *Definitions, classification, and epidemiology of obesity*. 2015.

Evgeny Putin, Polina Mamoshina, Alexander Aliper, Mikhail Korzinkin, Alexey Moskalev, Alexey Kolosov, Alexander Ostrovskiy, Charles Cantor, Jan Vijg, and Alex Zhavoronkov. Deep biomarkers of human aging: application of deep neural networks to biomarker development. *Aging (Albany NY)*, 8(5):1021, 2016.

Abeer Qannan, Yosra Bejaoui, Mahmoud Izadi, Noha A Yousri, Aleem Razzaq, Colette Christiansen, George M Martin, Jordana T Bell, Steve Horvath, Junko Oshima, Andre Megarbane, Johan Ericsson, Ehsan Pourkarimi, and Nady El Hajj. Accelerated epigenetic aging and DNA methylation alterations in Berardinelli-Seip congenital lipodystrophy. *Hum. Mol. Genet.*, 32(11):1826–1835, May 2023.

Zhen Qian, Yuancun Li, Zhiqiang Guan, Pi Guo, Ke Zheng, Yali Du, Shengjie Yin, Binyao Chen, Hongxi Wang, Jiao Jiang, Kunliang Qiu, and Mingzhi Zhang. Global, regional, and national burden of multiple sclerosis from 1990 to 2019: Findings of global burden of disease study 2019. *Front. Public Health*, 11: 1073278, February 2023.

Vardhman K Rakyan, Huriya Beyan, Thomas A Down, Mohammed I Hawa, Siarhei Maslau, Deeqo Aden, Antoine Daunay, Florence Busato, Charles A Mein, Burkhard Manfras, Kerith-Rae M Dias, Christopher G Bell, Jörg Tost, Bernhard O Boehm, Stephan Beck, and R David Leslie. Identification of type 1 diabetes-associated DNA methylation variable positions that precede disease diagnosis. *PLoS Genet.*, 7(9):e1002300, September 2011.

Bruno Ramos-Molina, Lidia Sánchez-Alcoholado, Amanda Cabrera-Mulero, Raul Lopez-Dominguez, Pedro Carmona-Saez, Eduardo Garcia-Fuentes, Isabel Moreno-Indias, and Francisco J Tinahones. Gut microbiota composition is associated with the global DNA methylation pattern in obesity. *Front. Genet.*, 10:613, July 2019.

Asma Rashki Kemmak, Aziz Rezapour, Reza Jahangiri, Shima Nikjoo, Hiro Farabi, and Samira Soleimanpour. Economic burden of osteoporosis in the world: A systematic review. *Med. J. Islam. Repub. Iran*, 34:154, November 2020.

Araz Rawshani, Naveed Sattar, Stefan Franzén, Aidin Rawshani, Andrew T Hattersley, Ann-Marie Svensson, Björn Eliasson, and Soffia Gudbjörnsdottir. Excess mortality and cardiovascular disease in young adults with type 1 diabetes in relation to age at onset: a nationwide, register-based cohort study. *Lancet*, 392(10146): 477–486, August 2018.

Lindsay M Reynolds, Jackson R Taylor, Jingzhong Ding, Kurt Lohman, Craig Johnson, David Siscovick, Gregory Burke, Wendy Post, Steven Shea, David R Jacobs, Jr, Hendrik Stunnenberg, Stephen B Kritchevsky, Ina Hoeschele, Charles E McCall, David Herrington, Russell P Tracy, and Yongmei Liu. Age-related variations in the methylome associated with gene expression in human monocytes and T cells. *Nat. Commun.*, 5(1): 5366, November 2014.

Brooke Rhead, Calliope Holingue, Michael Cole, Xiaorong Shao, Hong L Quach, Diana Quach, Khooshbu Shah, Elizabeth Sinclair, John Graf, Thomas Link, Ruby Harrison, Elior Rahmani, Eran Halperin, Wei Wang, Gary S Firestein, Lisa F Barcellos, and Lindsey A Criswell. Rheumatoid arthritis naive T cells share hypermethylation sites with synoviocytes. *Arthritis Rheumatol.*, 69(3):550–559, March 2017.

Janou A Y Roubroeks, Adam R Smith, Rebecca G Smith, Ehsan Pishva, Zina Ibrahim, Martina Sattlecker, Eilis J Hannon, Iwona Kłoszewska, Patrizia Mecocci, Hilkka Soininen, Magda Tsolaki, Bruno Vellas, Lars-Olof Wahlund, Dag Aarsland, Petroula Proitsi, Angela Hodges, Simon Lovestone, Stephen J Newhouse, Richard J B Dobson, Jonathan Mill, Daniël L A van den Hove, and Katie Lunnon. An epigenome-wide association study of alzheimer's disease blood highlights robust DNA hypermethylation in the HOXB6 gene. *Neurobiol. Aging*, 95:26–45, November 2020.

Paz L D Ruiz, Lei Chen, Jedidiah I Morton, Agus Salim, Bendix Carstensen, Edward W Gregg, Meda E Pavkov, Manel Mata-Cases, Didac Mauricio, Gregory A Nichols, Santa Pildava, Stephanie H Read, Sarah H Wild, Jonathan E Shaw, and Dianna J Magliano. Mortality trends in type 1 diabetes: a multicountry analysis of six population-based cohorts. *Diabetologia*, 65(6):964–972, June 2022.

Jarod Rutledge, Hamilton Oh, and Tony Wyss-Coray. Measuring biological age using omics data. *Nature Reviews Genetics*, 23(12):715–727, 2022.

Lisa Ruvuna and Akshay Sood. Epidemiology of chronic obstructive pulmonary disease. *Clin. Chest Med.*, 41 (3):315–327, September 2020.

Lotte Saaby, Tina Svenstrup Poulsen, Axel Cosmus Pyndt Diederichsen, Susanne Hosbond, Torben Bjerregaard Larsen, Henrik Schmidt, Oke Gerke, Jesper Hallas, Kristian Thygesen, and Hans Mickley. Mortality rate in type 2 myocardial infarction: observations from an unselected hospital cohort. *Am. J. Med.*, 127(4):295–302, April 2014.

Sanish Sathyan, Emmeline Ayers, Tina Gao, Erica F Weiss, Sofiya Milman, Joe Verghese, and Nir Barzilai. Plasma proteomic profile of age, health span, and all-cause mortality in older adults. *Aging Cell*, 19(11): e13250, 2020.

Mehmet Sayiner, Pegah Golabi, Maria Stepanova, Issah Younossi, Fatema Nader, Andrei Racila, and Zobair M Younossi. Primary biliary cholangitis in medicare population: The impact on mortality and resource use. *Hepatology*, 69(1):237–244, January 2019.

Franziska Schnabel, Uwe Kornak, and Bernd Wollnik. Premature aging disorders: A clinical and genetic compendium. *Clin. Genet.*, 99(1):3–28, January 2021.

M Seip and O Trygstad. Generalized lipodystrophy, congenital and acquired (lipoatrophy). *Acta Paediatr. Suppl.*, 413:2–28, June 1996.

Christian P Selinger and Rupert W Leong. Mortality from inflammatory bowel diseases. *Inflamm. Bowel Dis.*, 18(8):1566–1572, August 2012.

Eyal Shahar, Seungmin Lee, Joseph Kim, Sue Duval, Cheryl Barber, and Russell V Luepker. Hospitalized heart failure: rates and long-term mortality. *J. Card. Fail.*, 10(5):374–379, October 2004.

Adam C Sheka, Oyedele Adeyi, Julie Thompson, Bilal Hameed, Peter A Crawford, and Sayeed Ikramuddin. Nonalcoholic steatohepatitis: A review. *JAMA*, 323(12):1175–1183, March 2020.

C G Solomon and J E Manson. Obesity and mortality: a review of the epidemiologic data. *Am. J. Clin. Nutr.*, 66(4 Suppl):1044S–1050S, October 1997.

Carolina Soriano-Tárraga, Jordi Jiménez-Conde, Eva Giralt-Steinhauer, Marina Mola-Caminal, Rosa M Vivanco-Hidalgo, Angel Ois, Ana Rodríguez-Campello, Elisa Cuadrado-Godia, Sergi Sayols-Baixeras, Roberto Elosua, Jaume Roquer, and GENESTROKE Consortium. Epigenome-wide association study identifies TXNIP gene associated with type 2 diabetes mellitus and sustained hyperglycemia. *Hum. Mol. Genet.*, 25(3):609–619, February 2016.

K Sutton-Tyrrell, H G Alcorn, H Herzog, S F Kelsey, and L H Kuller. Morbidity, mortality, and antihypertensive treatment effects by extent of atherosclerosis in older adults with isolated systolic hypertension. *Stroke*, 26 (8):1319–1324, August 1995.

Weiyang Tao, Arno N Concepcion, Marieke Vianen, Anne C A Marijnissen, Floris P G J Lafeber, Timothy R D J Radstake, and Aridaman Pandit. Multiomics and machine learning accurately predict clinical response to adalimumab and etanercept therapy in patients with rheumatoid arthritis. *Arthritis Rheumatol.*, 73(2): 212–222, February 2021.

Marcello Tonelli, Natasha Wiebe, Bruce Culleton, Andrew House, Chris Rabbat, Mei Fok, Finlay McAlister, and Amit X Garg. Chronic kidney disease and mortality risk: a systematic review. *J. Am. Soc. Nephrol.*, 17 (7):2034–2047, July 2006.

Adam Trickey, Caroline A Sabin, Greer Burkholder, Heidi Crane, Antonella d'Arminio Monforte, Matthias Egger, M John Gill, Sophie Grabar, Jodie L Guest, Inma Jarrin, Fiona C Lampe, Niels Obel, Juliana M Reyes, Christoph Stephan, Timothy R Sterling, Ramon Teira, Giota Touloumi, Jan-Christian Wasmuth, Ferdinand Wit, Linda Wittkop, Robert Zangerle, Michael J Silverberg, Amy Justice, and Jonathan A C Sterne. Life expectancy after 2015 of adults with HIV on long-term antiretroviral therapy in europe and north america: a collaborative analysis of cohort studies. *Lancet HIV*, 10(5):e295–e307, May 2023.

Gaël Varoquaux and Veronika Cheplygina. Machine learning for medical imaging: methodological failures and recommendations for the future. *NPJ digital medicine*, 5(1):48, 2022.

Miri Varshavsky, Gil Harari, Benjamin Glaser, Yuval Dor, Ruth Shemer, and Tommy Kaplan. Accurate age prediction from blood using a small set of dna methylation sites and a cohort-based machine learning algorithm. *Cell Reports Methods*, 3(9), 2023.

N T Ventham, N A Kennedy, A T Adams, R Kalla, S Heath, K R O'Leary, H Drummond, IBD BIOM consortium, IBD CHARACTER consortium, D C Wilson, I G Gut, E R Nimmo, and J Satsangi. Integrative epigenome-wide analysis demonstrates that DNA methylation may mediate genetic risk in inflammatory bowel disease. *Nat. Commun.*, 7:13507, November 2016.

Laura Vidal-Bralo, Yolanda Lopez-Golan, and Antonio Gonzalez. Simplified assay for epigenetic age estimation in whole blood of adults. *Frontiers in genetics*, 7:126, 2016.

Allison W Willis, Mario Schootman, Nathan Kung, Bradley A Evanoff, Joel S Perlmutter, and Brad A Racette. Predictors of survival in patients with parkinson disease. *Arch. Neurol.*, 69(5):601–607, May 2012.

Alison K Wright, Evangelos Kontopantelis, Richard Emsley, Iain Buchan, Naveed Sattar, Martin K Rutter, and Darren M Ashcroft. Life expectancy and cause-specific mortality in type 2 diabetes: A population-based cohort study quantifying relationships in ethnic subgroups. *Diabetes Care*, 40(3):338–345, March 2017.

Sheng Hui Wu, Zhong Liu, and Suzanne C Ho. Metabolic syndrome and all-cause mortality: a meta-analysis of prospective cohort studies. *Eur. J. Epidemiol.*, 25(6):375–384, June 2010.

Xian Xia, Xingwei Chen, Gang Wu, Fang Li, Yiyang Wang, Yang Chen, Mingxu Chen, Xinyu Wang, Weiyang Chen, Bo Xian, et al. Three-dimensional facial-image analysis to predict heterogeneity of the human ageing rate and the impact of lifestyle. *Nature metabolism*, 2(9):946–957, 2020.

Xian Xia, Yiyang Wang, Zhengqing Yu, Jiawei Chen, and Jing-Dong J Han. Assessing the rate of aging to monitor aging itself. *Ageing Research Reviews*, 69:101350, 2021.

Ying Xian, Robert G Holloway, Wenqin Pan, and Eric D Peterson. Challenges in assessing hospital-level stroke mortality as a quality measure: comparison of ischemic, intracerebral hemorrhage, and total stroke mortality rates. *Stroke*, 43(6):1687–1690, June 2012.

Shiyu Xiao, Wenhui Xie, Yinghui Zhang, Lei Lei, and Yan Pan. Changing epidemiology of cirrhosis from 2010 to 2019: results from the global burden disease study 2019. *Ann. Med.*, 55(2):2252326, 2023.

Jane Xu, Ching S Wan, Kiriakos Ktoris, Esmee M Reijnierse, and Andrea B Maier. Sarcopenia is associated with mortality in adults: A systematic review and meta-analysis. *Gerontology*, 68(4):361–376, 2022.

Kejun Ying, Seth Paulson, Alec Eames, Alexander Tyshkovskiy, Siyuan Li, Martin Perez-Guevara, Mehrnoosh Emamifar, Maximiliano Casas Martínez, Dayoon Kwon, Anna Kosheleva, et al. A unified framework for systematic curation and evaluation of aging biomarkers. *bioRxiv*, pp. 2023–12, 2023.

Kejun Ying, Hanna Liu, Andrei E Tarkhov, Marie C Sadler, Ake T Lu, Mahdi Moqri, Steve Horvath, Zoltán Kutalik, Xia Shen, and Vadim N Gladyshev. Causality-enriched epigenetic age uncouples damage and adaptation. *Nature Aging*, pp. 1–16, 2024.

Zobair Younossi, Frank Tacke, Marco Arrese, Barjesh Chander Sharma, Ibrahim Mostafa, Elisabetta Bugianesi, Vincent Wai-Sun Wong, Yusuf Yilmaz, Jacob George, Jiangao Fan, and Miriam B Vos. Global perspectives on nonalcoholic fatty liver disease and nonalcoholic steatohepatitis. *Hepatology*, 69(6):2672–2682, June 2019.

Ming Yu, William D Hazelton, Georg E Luebeck, and William M Grady. Epigenetic aging: more than just a clock when it comes to cancer. *Cancer research*, 80(3):367–374, 2020.

Nicholas Chak Lam Yung, Corine Sau Man Wong, Joe Kwun Nam Chan, Eric Yu Hai Chen, and Wing Chung Chang. Excess mortality and life-years lost in people with schizophrenia and other non-affective psychoses: an 11-year population-based cohort study. *Schizophrenia Bulletin*, 47(2):474–484, 2021.

Qian Zhang, Costanza L Vallerga, Rosie M Walker, Tian Lin, Anjali K Henders, Grant W Montgomery, Ji He, Dongsheng Fan, Javed Fowdar, Martin Kennedy, et al. Improved precision of epigenetic clock estimates across tissues and its implication for biological ageing. *Genome medicine*, 11:1–11, 2019.

Xinyu Zhang, Amy C Justice, Ying Hu, Zuoheng Wang, Hongyu Zhao, Guilin Wang, Eric O Johnson, Brinda Emu, Richard E Sutton, John H Krystal, and Ke Xu. Epigenome-wide differential DNA methylation between HIV-infected and uninfected individuals. *Epigenetics*, 11(10):750–760, October 2016.

Xinyu Zhang, Ying Hu, Amy C Justice, Boyang Li, Zuoheng Wang, Hongyu Zhao, John H Krystal, and Ke Xu. DNA methylation signatures of illicit drug injection and hepatitis C are associated with HIV frailty. *Nat. Commun.*, 8(1):2243, December 2017a.

Xinyu Zhang, Ying Hu, Bradley E Aouizerat, Gang Peng, Vincent C Marconi, Michael J Corley, Todd Hulgan, Kendall J Bryant, Hongyu Zhao, John H Krystal, Amy C Justice, and Ke Xu. Machine learning selected smoking-associated DNA methylation signatures that predict HIV prognosis and mortality. *Clin. Epigenetics*, 10(1):155, December 2018.

Yuqing Zhang, Na Lu, Christine Peloquin, Maureen Dubreuil, Tuhina Neogi, J Antonio Aviña-Zubieta, Sharan K Rai, and Hyon K Choi. Improved survival in rheumatoid arthritis: a general population-based cohort study. *Ann. Rheum. Dis.*, 76(2):408–413, February 2017b.

Alex Zhavoronkov, Kirill Kochetov, Peter Diamandis, and Maria Mitina. Psychoage and subjage: development of deep markers of psychological and subjective age using artificial intelligence. *Aging (Albany NY)*, 12(23):23548, 2020.

Rongrong Zhu, Shan Zhou, Liang Xia, and Xiaoming Bao. Incidence, morbidity and years lived with disability due to type 2 diabetes mellitus in 204 countries and territories: Trends from 1990 to 2019. *Front. Endocrinol. (Lausanne)*, 13:905538, July 2022a.

Rongrong Zhu, Shan Zhou, Liang Xia, and Xiaoming Bao. Incidence, morbidity and years lived with disability due to type 2 diabetes mellitus in 204 countries and territories: Trends from 1990 to 2019. *Front. Endocrinol. (Lausanne)*, 13:905538, July 2022b.

