# OpenReview forum: "ComputAgeBench: Epigenetic Aging Clocks Benchmark"
_ICLR.cc/2025/Conference — Submitted to ICLR 2025_

### Official Review · Reviewer_jKRR · 2024-11-02

**Soundness:** 3
**Presentation:** 3
**Contribution:** 2
**Rating:** 6
**Confidence:** 5

**Summary:**

The paper proposes ComputAgeBench, a unified framework and benchmark dataset for evaluating epigenetic aging clocks, which are predictive models for estimating biological age based on DNA methylation data.  The framework aggregates 66 public datasets covering 19 aging-accelerating conditions, along with 13 published epigenetic clock models, to assess model performance consistently across a standardized dataset. The methodology incorporates rigorous evaluation criteria to test each model’s ability to distinguish between healthy individuals and those with accelerated aging conditions.

**Strengths:**

### Strengths

 The paper is clear and well-written, providing a solid foundation for its contributions. It presents a unified framework for evaluating epigenetic aging clocks, covering both first- and second-generation clocks. By introducing a benchmark dataset, the authors enable comprehensive testing of multiple epigenetic clock methods.

This work has potential to significantly impact the field of biological aging, as it offers a standardized dataset that can facilitate consistent evaluation across various epigenetic clock methods. Such a resource will likely streamline method comparison and improve reliability in aging research.

**Weaknesses:**

In reviewing the proposed benchmark in this paper, several key areas for improvement have emerged, particularly concerning data diversity, balance, and bias.



### Weaknesses



1. **Limited Report on Data Diversity**: The paper lacks adequate details on demographic and biological diversity, such as age, ethnicity, and health variations. Including these would improve the dataset's representativeness for broader applications.



2. **Data Balance and Bias**: The authors do not address balance across categories (e.g., AACs vs. healthy controls) or potential sampling biases. This oversight may skew benchmarking results and limit generalizability.



3. **Absence of Bias Mitigation**: No strategies are mentioned to detect or reduce dataset biases, which is crucial for fair benchmarking in aging prediction models, where demographic factors can affect DNA methylation patterns and model performance. Additional evaluation metrics for fairness would  increase the strength of this benchmark.



4. **Put Together Publicly Available Dataset**: The proposed dataset, to my understanding,  is a collection of existing publicly available datasets. The authors do not present to the research community a new benchmarking dataset, they rather collect existing datasets that they put together with a published harmonization technique.

The fact that the datasets already exist publicly, reduces the novelty of the benchmark. However, I cannot ignore that putting together 66 datasets  into a single dataset is a contribution that would facitilitate the comparison of epigenetic clock methods.

**Questions:**

### Questions for the Authors

In evaluating the dataset and methodology presented, several questions arose that could help clarify the dataset’s potential applications and limitations.

1. **Applicability for Method Development**: Can this dataset be effectively used for developing new methods on epigenetic aging clocks, or is it primarily intended for benchmarking and evaluation? Are there features or structures in the dataset that support novel method exploration?

2. **Data Diversity and Representativeness**: How does the dataset account for demographic and biological diversity? Could the authors provide more details on the inclusion criteria to ensure the dataset is representative of a broad population?

3. **Addressing Balance and Bias**: Were any steps taken to balance the dataset across aging-accelerating conditions (AACs) and healthy controls, or to mitigate known biases in the sample selection process?

---

> ### Author Response · Authors · 2024-11-20
> **Rebuttal by Authors**
>
> **Q1: ...developing new methods on epigenetic aging clocks...**
>
> The dataset, gathered for benchmarking, is also suitable for differential methylation analysis, clustering, and developing epigenetic clocks. It enables exploration of novel clocks **targeting CpG sites distinguishing aging-accelerating conditions (AACs) from healthy controls (HCs)**. Spanning **19 AACs, both genders, and a 90-year age range**, it offers valuable opportunities for clock research. While the primary aim was to validate clocks for estimating accelerated aging, not training them, the HC cohort can still support clock training, complementing ongoing open-access efforts [R1].
>
> Reference:
>
> R1. Ying, K., ... & Gladyshev, V. N. (2023). A Unified Framework for Systematic Curation and Evaluation of Aging Biomarkers. bioRxiv, 2023-12.
>
> **Q2: How does the dataset account for demographic and biological diversity?...**
>
> Thank you for your comment. While collecting the dataset, we strived to find balance between data quantity and diversity. Because the published sets of DNA methylation (DNAm) data come from different studies with varying goals, they often lack a thorough annotation of patient health and other conditions. By aggregating as much data sources as possible, without excluding any genders or ethnicities, we presented the most representative collection of human DNAm profiles in health and age-related disease. The **details on age and gender distribution** within AACs and HCs per dataset are provided in **Appendix figures A1, A2** (Ref. section A.8). Clearly, there is room for improvement in terms of data sources, which we outlined explicitly in section **A.1**. Additionally, we have calculated the following Table R1 with sample counts, showing that the majority of data is unlabeled by ethnicity (and sometimes gender), even though we did our best to find quality data.
>
> Table R1. Sample counts across by ethnicities and genders (M=male, F=Female, U=Unknown) in the aggregated dataset.
> | Ethnicity  | Gender | AAC  | HC  |
> |------------|--------|------|-----|
> | Unknown    | M      | 1620 | 603 |
> |            | F      | 1476 | 849 |
> |            | U      | 957  | 417 |
> | White      | M      | 1884 | 364 |
> |            | F      | 577  | 398 |
> | Black      | M      | 699  | 69  |
> |            | F      | 36   | 17  |
> |            | U      | 3    | 1   |
> | Hispanic   | M      | 97   | 55  |
> |            | F      | 50   | 48  |
> | Asian      | M      | 3    | 4   |
> |            | F      | 10   | 9   |
> | Other      | M      | 159  | 14  |
> |            | F      | 3    | 0   |
>
> **Q3: ...to balance the dataset across ... AACs and healthy controls, or to mitigate known biases in the sample selection process?**
>
> To ensure that the datasets were not biased towards specific AACs, we first defined a set of criteria for AACs and datasets, targeting **major human organ systems** and then acquired 66 datasets, covering 19 of the 32 identified conditions (Ref. Table A2 for details). We aimed to assemble a comprehensive dataset to validate aging clocks, while **mitigating any bias in sample selection through the diversity and large number of studies** included. To prevent scenarios where a given aging clock might favor better predictions for a particular class of conditions (e.g., ISD), we present a **decomposition of the clock’s AA2 and AA1 scores**, as shown in Figure 3E,F. Additionally, we address the balance across categories by presenting sample distributions across conditions, ages, and demographic groups in Appendix Figures A1 and A2. These decompositions effectively **prevent misinterpretations and reduce potential clock bias caused by dataset imbalances across the AACs**.
>
> **Q4: ...datasets already exist publicly, reduces the novelty of the benchmark. However, ... putting together 66 datasets ... is a contribution...**
>
> Indeed, gathering and curating a new dataset of this size is a complex and a time-intensive endeavor (took us three years), because there are no options for an immediate selection of data relevant to the clock research. Collecting a large number of new DNAm samples from humans is increasingly difficult due to the high costs, significant time investment, and ethical challenges. This is also the reason why many clock-related papers **use chaotically varying sets of data**: everyone uses what they could find, with different teams managing to pre-process and split the data differently. Unfortunately, the rationale behind including or excluding data on a particular health/disease condition is rarely articulated as well.
>
> Hence, by introducing **a clearly stated methodology for conditions selection** and by consolidating open-access datasets, we aspire to remove these barriers and to provide the research community with a valuable, easily accessible resource to accelerate the validation of aging clocks in a standardized fashion.

---

> > ### Comment · Reviewer_jKRR · 2024-11-22
> > **Response**
> >
> > Thank you for addressing all my questions. I do believe that this line of work will have an important impact on epigenetic clock research. Thus, I raise my score to 6 and I recommend this paper for acceptance. Best of luck!

---

> > > ### Author Response · Authors · 2024-11-28
> > >
> > > We sincerely thank you for your thoughtful feedback, which enhanced the quality of our manuscript, and for increasing your score.

---

### Official Review · Reviewer_3vvy · 2024-11-03

**Soundness:** 2
**Presentation:** 3
**Contribution:** 2
**Rating:** 5
**Confidence:** 4

**Summary:**

The authors present a benchmark study where they contrast different computational methods, namely aging clocks, for inferring biological age from epigenetics (methylation) data. A corpus of datasets relevant for the benchmark was built through a systematic search, and it is provided as a resource. Finally, the evaluation was performed on four different tasks, devised in such a way to capture different aspects of aging clocks' performances.

**Strengths:**

The benchmark is well structured: (i) a variety of datasets and methods are included, and (ii) the tasks upon which the methods are evaluated are well defined and relevant for the domain. Furthermore, such type of benchmarks are quite timely, due to a continuously growing list of available aging clocks.

**Weaknesses:**

My main criticism is that the paper is only marginally relevant with respect to the topics of the conference. Inferring the biological age of an individual can hardly be considered as learning representations. The machine learning methods used for deriving aging clocks are very well known and established, thus lacking novelty. The tasks presented in the paper to assess the clocks' performances are not totally novel, as the authors themselves point out in section 2.2.

From a technical point of view, an important aspect that the paper does not address is preprocessing. Several normalization methods exist for methylation data, and their impact to downstream analysis is well documented (see for example Teschendorff et al. 2013). A robust benchmark should try to evaluate the effect of different normalization methods on aging clock performances.

A minor issue the authors may want to consider: the long list of reference at page 6 could be placed in the appendix, to ease reading

Andrew E. Teschendorff, Francesco Marabita, Matthias Lechner, Thomas Bartlett, Jesper Tegner, David Gomez-Cabrero, Stephan Beck, A beta-mixture quantile normalization method for correcting probe design bias in Illumina Infinium 450 k DNA methylation data, Bioinformatics, Volume 29, Issue 2, January 2013, Pages 189–196,

**Questions:**

I would like to ask the authors to address the two main criticisms I listed in the "weaknesses" section:
- Overall, the opinion of this reviewer is that while the work has undoubtedly merit, it would be better suited for a forum more specific to biological age and aging clocks.
- Regarding the normalization of methylation data, I would invite the authors to at least discuss whether the preprocessing of the included datasets match the recommended preprocessing of each aging clock (if any).

---

> ### Author Response · Authors · 2024-11-20
> **Rebuttal by Authors**
>
> **"Q1: ..better suited for a forum more specific to biological age.."**
>
> Although we hear the concern, we respectfully argue that the biological age, as presented in our study, aligns closely with the goals of representation learning, because the community looks for an **interpretable univariate representation of complex biomarker data**. Biological age encapsulates a high-dimensional array of biomarkers into a **single, accessible latent variable**. This process is achieved by training models under specific assumptions that have evolved across the generations of aging clocks, similar to how representation learning frameworks have refined their assumptions to improve the quality and relevance of latent representations in other biologically-relevant tasks presented at ICLR previously [R1, R2, R3].
>
> Furthermore, the utility of biological age extends into numerous downstream applications, such as predicting all-cause mortality, estimating multi-morbidity risk, and evaluating general health status. These applications literally highlight its role as an **interpretable and practically valuable representation** with real-world impact [R4, R5]. In this way, biological age enriches traditional representation learning by adding interpretability and ease of use, bridging machine learning and healthcare.
> Regarding the tasks proposed in our benchmark, we emphasize that the task selection is inspired by integrating the established practices of aging clock evaluation found in numerous studies. We employ a clear, interpretable approach for testing aging clocks, which is relevant to a broader research community in data and life sciences.
>
> **Q2: ..normalization..**
>
> During the data pre-processing step of our study, we were primarily oriented at **the most common use case of applying aging clocks** by anyone other than the clockmaker. That is, when a researcher collects and processes their data in a way they find the most appropriate (for data, not for the aging clocks they plan to use), and then apply an aging clock model **trained on other data, already pre-processed** by the clock authors. As there is no gold standard for DNA methylation pre-processing, **each research group carries out their own pre-processing that does not necessarily match the pre-processing pipeline used for training the clock model**. For example, this is almost always the case when older clock models, such as Horvath2013 or Hannum2013, are applied to recently acquired data.
>
> Therefore, so as to retain this typical workflow and not to put any clock model into advantage by choosing the same pre-processing that matches its own pipeline for every dataset, we decided to include already pre-processed datasets. In doing so, we also relied on two existing papers. First, compiling already pre-processed datasets without performing the same pre-processing for all of them was done in [R6], another notable effort in the aging clock community. Second, we were also encouraged by a recent paper by Varshavsky et al. [R7] who managed to **create an accurate clock model** by combining several blood datasets—without any additional normalization or correction procedure, **using already pre-processed data** from previous studies (some of which are included in our dataset as well), and thus demonstrating that the **between-dataset normalization is not critical for this type of data**.
> We discussed our reasoning in the Section 3.3 of our Benchmarking Methodology chapter and in the Appendix section A.9, and we thank you for pointing out the potential misunderstanding that we will clarify in the revised text.
>
> **Q3: A minor issue...**
>
> Admittedly, we had considered removing this list in the Appendix, but eventually kept it in the Methodology section so that all the respective studies of DNA methylation profiling would be cited in the main references list. However, this issue is indeed minor, and we are open to replacing the references in the revised version.
>
> **References**
>
> R1 Marin, F. I., et al. Bend: Benchmarking dna language models on biologically meaningful tasks. // ICLR 2024
>
> R2 Pandeva T., Forré P. Multi-View Independent Component Analysis for Omics Data Integration //2023 ICLR
>
> R3 Zhou Z. et al. Dnabert-2: Efficient foundation model and benchmark for multi-species genome //arXiv preprint arXiv:2306.15006. – 2023.
>
> R4 Pyrkov T. V. et al. Longitudinal analysis of blood markers reveals progressive loss of resilience and predicts human lifespan limit //Nature communications. – 2021.
>
> R5 Pierson E. et al. Inferring multidimensional rates of aging from cross-sectional data //The 22nd International Conference on Artificial Intelligence and Statistics. – PMLR, 2019.
>
> R6 Ying, K., et al. (2023). A Unified Framework for Systematic Curation and Evaluation of Aging Biomarkers. bioRxiv, 2023-12.
>
> R7 Varshavsky, M., et al. (2023). Accurate age prediction from blood using a small set of DNA methylation sites and a cohort-based machine learning algorithm. Cell Reports Methods.

---

> > ### Author Response · Authors · 2024-11-26
> >
> > Dear Reviewer,
> >
> > Please kindly let us know if you are satisfied with our responses and if we can do anything else to support our work.

---

> > > ### Comment · Reviewer_3vvy · 2024-11-27
> > >
> > > I would like to thanks the authors for their effort in replying to my comments. Unfortunately, I still have concerns regarding this submission.
> > >
> > > First, I still consider this work only marginally relevant for the ICLR conference. I do appreciate the clarification offered by the authors regarding the latent nature of biological age. However, the major contribution of the paper remains its curated selection of methylation datasets suitable for age clocks' training / validation, which again I think would be better suitable for a more specialized journal or conference.
> > >
> > > Regarding data normalization, I disagree with the authors' statement "to retain this typical workflow and not to put any clock model into advantage by choosing the same pre-processing that matches its own pipeline for every dataset". The fact that the "typical workflow" seems to be "preprocessing their own data while ignoring the preprocessing used during the creation of the clock", does not mean that this workflow is correct. Rather, this shows that there is a need of evaluating the most suitable normalization approach for each age clock, so that researchers can make an informed choice when they pair a preprocessing algorithm and an age clock. I understand that such a onerous task might be outside of the scope of this paper, however I would urge the authors to elaborate more on this point, better underlying that the effect of preprocessing on biological age estimation is still an under-investigated topic.

---

> > > > ### Author Response · Authors · 2024-11-28
> > > >
> > > > We thank you for elaborating your concerns!
> > > >
> > > > Respectfully, we argue that the foremost contribution of our work was not a collection of datasets. Given time and effort, the curation of such a stack is tedious, but trivial. Taken individually, the criteria for selecting conditions and datasets, and the tasks that we used are also not novel. But, more importantly, our work for the first time aggregates them all in a **coherent, rigorous, and clinically relevant methodology to compare epigenetic clock models** and, yes, we also provided the open-access datasets that are fit specifically for this methodology.
> > > >
> > > > Currently, our proposal is **the only effort to define conditions that the clocks should be compared by** (in cases when epigenetic data combined with mortality is not publicly available, which is always, unfortunately). Even in the best cases, the authors of a clock paper choose their benchmarking datasets ad hoc (when a model is already built) and without openly clarifying why they limited themselves to that specific combination. This leads to a situation, in which we, as observers, **cannot verify whether their clocks can actually be trusted for estimating biological age**, because we cannot readily test them on any other severe condition which is expected to impact a person's biological age dramatically. By our work, we let any researcher validate any model on the largest collection of epigenetic data in life-shortening conditions.
> > > >
> > > > Concerning data processing, we would surely love to see that the newer datasets are always well-paired with older clock models, however this task is of lesser concern, because the main issue with aging clock nowadays is not lack of appropriate normalization. As we mentioned in the earlier reply, Varshavsky et al. (2023) had managed to create accurate clocks by combining several datasets without any processing at all. **More crucial and far less established** is the methodology of validating a clock's predictive ability to notice significant changes in patient's health, thus being a good indicator of biological age. Without such methodology, no processing can help.
> > > >
> > > > Similarly to what we commented in an other reply to reviewer XgQd, we do not expect that our approach will become an immutable canon carved in stone. Clinically relevant methodology for validating *latent biomarkers of aging* is necessary, but it should arise from **a scientific discussion between clinicians, aging biologists, and those proficient at learning representations**. Therefore, by presenting it to the ML community, we strive to pave the road for the extended network of researchers to attract their attention to the task of modeling aging with epigenetic data and engage in this productive discussion.

---

### Official Review · Reviewer_XgQd · 2024-11-11

**Soundness:** 3
**Presentation:** 4
**Contribution:** 2
**Rating:** 6
**Confidence:** 4

**Summary:**

This paper benchmarks 13 different published biological clock models using a standardized test dataset that they compiled from more than 50 different publicly available studies. While no ground truth data is available for biological age (as it is a latent factor) or for age at death (as this data often isn’t published), the authors offer 4 compelling metrics by which to score the models accuracy and robustness. This paper presents a resource to the community in terms of a newly published benchmarking dataset, well-motivated metrics, and ratings for the current state of the art clock models. The paper also appropriately outlines limitations, such as the fact that some datasets had poor performance across all models, raising questions about dataset shift and for what kinds of data the clocks can be expected to make sound predictions. I believe this paper will help generate scientific discussion and progress in the aging clocks research community.

**Strengths:**

- This paper is written very clearly, and did a great job walking the reader through the background to the problem, definitions of biological age, and different kinds of biological clock models. It’s graphics are informative, clear, and aesthetic. Truly a pleasure to read!
- Provides colab notebook for reproducibility
- I believe this paper will be significant to those in the biological clocks community. It is a benchmarking paper, so while it doesn't offer a new methodology itself, it does offer original tasks/metrics for assessing the performance of these models (I think they are original, I asked for clarification in the questions section) and a standardized benchmarking dataset (I asked for clarity to confirm it will in fact be published along with this paper)

**Weaknesses:**

- I was disappointed that the clock models weren't all re-trained on a standardized training dataset. Without standardizing the training data, it is impossible to know whether the methodology of the clock or the training data it used are contributing to better/worse performance. This insight would be critical to the community in improving clock methodologies going forward.
- The way that the authors chose to combine benchmarks in the cumulative score requires more justification. I am not sure why the different metrics should affect each other's weights so much. A simple sum, or weighted sum, of the four variables might be more appropriate if stronger justification is not supplied.
- Requires clarification: on the one hand, authors write "Clearly, the first task [AA1] provides a more rigorous way to test aging clocks [compared to AA2]" on the other hand, they write "The most rigorous of the four, AA2 task demonstrates..."
- Your description of the biomarker paradox could be improved. When I first read your description, I was left with questions. I had trouble finding more info on the "paradox of biomarkers" using the papers you cited (possibly due to paywall issues, I couldn't see the full articles), but you might consider adding this reference _Sluiskes, Marije H., et al. "Clarifying the biological and statistical assumptions of cross-sectional biological age predictors: an elaborate illustration using synthetic and real data." BMC Medical Research Methodology 24.1 (2024): 58._ as their explanation made me fully understand the problem, namely that "a (bio)marker that perfectly correlates with chronological age is useless in estimating biological age... in principle a nearly perfect chronological age predictor can be developed, as long as the sample size is large enough [35]. In such a case all signal related to biological aging would be lost."

More broadly, while I really enjoyed the paper, I am not sure it is a great fit for the ICLR community, as this model is a predictive regression model and not in the space of representation learning.

**Questions:**

- Will you make your benchmarking dataset publicly available? Can you please add a link to it in your manuscript? I view this benchmarking dataset as a significant portion of your contribution in this work.
- Can you please confirm that your evaluation tasks/metrics are original, and add citations if not?
- Can you make a case for why the paper is a strong fit for ICLR, despite not truly being in the representation learning space?

---

> ### Author Response · Authors · 2024-11-20
> **Rebuttal by Authors**
>
> **Q1: Will you make your benchmarking dataset publicly available?**
>
> Yes, we will definitely make it fully available, and the links to both the dataset and the code required to reproduce all figures are already prepared, but are omitted in the anonymized version of this submission.
>
> **Q2: Can you please confirm that your evaluation tasks/metrics are original, and add citations if not?** & **W2: ..cumulative score requires more justification**
>
> We write in the article: “Two approaches we propose as essential tasks in our benchmark entail related prior art. For example, Porter et al. [41] and Mei et al. [34] used one-sample or two-sample aging acceleration tests for clock validation”, demonstrating that the AA2 (see [R1, R2]) and AA1 (see [R3]) tasks were well-established and actively utilized previously. Hence, constructing a score based on these approaches for clock validation was a natural choice. Furthermore, calculating the median absolute error (as in [R4, R5]) to measure the accuracy of chronological age prediction is also a well-known approach.
>
> The reason a weighted sum of the four metrics is not applicable in our case is that only the AA2 and AA1 tasks are essential for testing clock validity, as outlined in the requirements provided in the Background section. The other two tasks serve as auxiliary measures, providing additional insights into the degree of clock calibration to chronological age.
>
> Please kindly refer to our response to Reviewer PX5Y Q2 for additional details regarding the rationale behind the design of the BenchScore metric.
>
> **Q3: Can you make a case for why the paper is a strong fit for ICLR, despite not truly being in the representation learning space?**
>
> Please kindly refer to our response to Reviewer 3vvy Q1, where we explain why the biological age is indeed a learnable representation. Think of the task of BA prediction that has no ground truth. It bears similarities with **unsupervised learning task** when the latent representation (BA) is obtained within a particular training procedure. For example, in the 1st generation clocks (or even unsupervised clocks), the BA cannot be verified with a ground truth – and yet – these clocks are still used in downstream clinical tasks. **Likewise, a significant portion of ICLR submissions is concerned with tasks that involve unknown targets, metrics, and data that lack annotation**. That is why we believe that the ML community is the best place where aging researchers could seek help from.
>
> **W1: ...weren't all re-trained on a standardized training dataset...**
>
> Indeed, the standardized training data are important, but often missing from clock studies, with the Biolearn effort [R1] being one of the few sources to provide it. Taking pre-trained models was exactly the point of doing our comparison, because **most published aging clock models employ the same architecture** (most often linear regression-based). **The outcomes are unique only thanks to their training data**. Therefore, re-training clock models on a single training dataset would create a completely novel clock that would have little in common with the published ones. Moreover, the **second-generation clocks rely on data combining mortality and DNAm values**, which, as noted in our manuscript, are either **unavailable or restricted** due to ethical concerns.
>
> **W3: Requires clarification: ..."Clearly, the first task [AA1] provides a more rigorous way to test aging clocks [compared to AA2]"...**
>
> Thank you for highlighting a potential misunderstanding. We will clarify by mentioning the tasks explicitly: “Clearly, the first task (AA2) provides a more rigorous way to test aging clocks (compared to AA1) …”. AA2 is more rigorous because it compares predictions between diseased and healthy patients, accounting for possible age acceleration due to batch effects, rather than comparing diseased patients to zero acceleration. This ensures that clocks systematically predicting accelerated ages for healthy subjects gain no advantage in AA2, but might appear successful in AA1. To address this, we penalize the AA1 score for prediction bias, which justifies the need for a cumulative benchmarking score.
>
>
> References:
>
> R1 Mei X et al. Fail-tests of DNA methylation clocks, and development of a noise barometer for measuring epigenetic pressure of aging and disease. Aging (Albany NY), 2023.
>
> R2 Ying K et al. Causality-enriched epigenetic age uncouples damage and adaptation. Nature Aging, 2024.
>
> R3 Porter HL et al. (2021). Many chronological aging clocks can be found throughout the epigenome: Implications for quantifying biological aging. Aging cell, 20(11), e13492.
>
> R4 Horvath S (2013). DNA methylation age of human tissues and cell types. Genome biology, 14, 1-20.
>
> R5 Thompson MJ et al. A multi-tissue full lifespan epigenetic clock for mice. Aging (Albany NY), 10(10), 2832.
>
> R6 Ying K et al. (2023). A Unified Framework for Systematic Curation and Evaluation of Aging Biomarkers. bioRxiv, 2023-12.

---

> > ### Author Response · Authors · 2024-11-26
> >
> > Dear Reviewer,
> >
> > Please kindly let us know if you are satisfied with our responses and if we can do anything else to support our work.

---

> > ### Comment · Reviewer_XgQd · 2024-11-27
> > **Response to Rebuttal**
> >
> > Thank you for taking the time to reply to the Reviewer comments and adding clarity. After reflecting on the paper and the authors' response to why it's a fit for ICLR, I maintain my position that while it is important work, it does not seem like a strong fit for ICLR. The models being compared are regression models, with the main difference between them being the data they were trained on. The major contribution of this paper is its benchmarking dataset (as the metrics are not novel), and while datasets & benchmarks papers are invited as ICLR submissions, this paper is really comparing how different training datasets affect performance rather than different machine learning models.

---

> ### Author Response · Authors · 2024-11-27
>
> We sincerely thank you for your additional reflection on our work and we hear your concern!
>
> First, the major contribution that we strived to make was not the dataset itself. It was an **explicit methodology of selecting the specific conditions and datasets**, based on clear, clinically relevant assumptions about how clocks should behave if we want them to be truly indicative of a person's biological age. The main problem with epigenetic aging clocks now, in our opinion, is that **there is no consensus way that they can be validated with using open-source data**, as biological age is a latent variable. By presenting our benchmark, we are, for the first time, providing researchers with a means of reliably comparing clock models, not just by how well they predict chronological age, but by how well they can perform in conditions that are proven to substantially decrease life expectancy. Our approach may not become consensus in the end, but, currently, **it is the only one that exists for omics data**, and we hope it will generate fruitful discussion in the field.
>
> Second, while most aging clocks indeed rely on linear models, their training methodologies differ significantly, with first-generation clocks predicting chronological age and second-generation clocks predicting mortality using Cox proportional hazard models, requiring differing assumptions about the biological age *as a learnt representation*.
>
> After three years of curating in-human data for our unifying benchmark, we see a critical need for ML expertise to develop robust biological age predictors, and we are well aware of the expectations at traditional ML conferences. It is to bridge this gap that we created our fully open-access benchmarking approach and repository. We hope that the optimal longevity markers will emerge through collaboration between the biological and ML communities.
>
> Lastly, please kindly note a trend in ICLR and other top conferences to accept similar works:
>
> -- Marin, et al. “Bend: Benchmarking dna language models on biologically meaningful tasks”, **ICLR 2024**.
>
> -- Sihag, et al. “Explainable brain age prediction using covariance neural networks”, **NeurIPS 2023**.
>
> -- Pandeva T., Forré P. “Multi-View Independent Component Analysis for Omics Data Integration”, **ICLR 2023**.
>
> -- Zhou Z. et al. “DNABERT-2: Efficient Foundation Model and Benchmark For Multi-Species Genomes”, **ICLR 2024**.
>
> -- Weinberger, E., & Lee, S. I. “A deep generative model of single-cell methylomic data”, **NeurIPS 2023** (GenBio Workshop).

---

### Official Review · Reviewer_PX5Y · 2024-11-11

**Soundness:** 3
**Presentation:** 3
**Contribution:** 3
**Rating:** 8
**Confidence:** 3

**Summary:**

The author introduces a benchmark designed to evaluate models of the epigenetic aging clock. The benchmark includes 66 datasets containing DNA methylation data that meet specific conditions and corresponding metadata, with a total sample size of 10,410. Four tasks are proposed to assess the models’ ability to distinguish between healthy individuals(HC) and age-accelerating conditions(ACC). Results of these four tests are summarized into Cumulative Benchmarking Score. The benchmark framework also includes 13 previously published models results.

**Strengths:**

The author critiques previous benchmarks for being either small in scale, limited to predicting chronological age, lacking standardized datasets, comparing only a limited number of models, or relying on mortality and disease data that have restricted access.

The proposed benchmark seems address all of these limitations. Derived from publicly accessible data, it includes processing of data from both age accelerating condition (ACC) and healthy control (HC) groups to test model’s ability to distinguish between these conditions. Diseases with ACC are well considered. The benchmark includes 4 well-defined tasks with a summary score and evaluates 13 previously published models.

**Weaknesses:**

The paper is well-written and comprehensive overall, but several technical points need further clarification:

1. The selection of metrics for benchmark tasks requires more justification. Specifically, why do tasks 2, 3, and 4 report median instead of the mean? Additionally, task 4 mentions the "presence of covariate shift," but this shift is not clearly explained. Could the authors specify the covariate shift further ?

2. The rationale behind the summary benchmark score requires further explanation. Why was this scoring method chosen, and what are its advantages? Also, what does "positive bias" refer to in this context? In the Results section, it is stated that $S_{AA1}$ is adjusted by a ratio to penalize prediction bias, yet this concept of prediction bias remains unexplained. Further clarification on what prediction bias entails here would be beneficial.

3. It appears that plots C and D in Figure 3 may be incorrectly presented. Plot D should likely represent $Med(|\Delta|)$ rather than $Med(\Delta)$, as all points are above the diagonal. Please clarify if this is a mislabeling or if I have misunderstood the data shown.

**Questions:**

Please see my questions in the above weakness section.

---

> ### Author Response · Authors · 2024-11-20
> **Rebuttal by Authors**
>
> **Q1: The selection of metrics ... "presence of covariate shift," but this shift is not clearly explained.**
>
> First, we want to thank you for pointing out a typo: in the description of the AA1 task (Section 3.5), we indeed meant “...mean aging acceleration…” instead of “median,” as the applied Student’s t-test is used to determine whether the difference in means between two groups is significant. We will correct this typo in the revised version of the text.
>
> Second, for tasks 3 and 4, the choice of the median is more appropriate due to its **robustness to outliers** compared to the mean. Thus, simplicity and robustness are the two primary reasons for favoring the median.
>
> Third, covariate shift, also referred to as a batch effect in bioinformatics, denotes the **shift between the distributions of covariates in two datasets**. For instance, the distribution of methylation values for a given CpG site could be centered around 0.45 in one dataset and around 0.55 in another—a common scenario in DNA methylation and other types of omics data. To evaluate the robustness of a given aging clock model to covariate shift (batch effect) between the original clock training dataset and datasets from the proposed benchmark, we introduced a **prediction bias task**. In this task, we calculate the median age acceleration, which reflects the **systematic shift in clock predictions caused by differences between datasets**. We will add explicit clarifications for the covariate shift in the revised version of the manuscript.
>
> **Q2: The rationale behind the summary benchmark score..**
>
> While designing our metric, we aimed for **simplicity and interpretability**. At the same time, we sought to include more data in the benchmark to address the data scarcity caused by the underrepresentation of certain AACs. In the simplest case, we could sum up the AA2 and AA1 scores; however, this approach would be unfair. **Clocks exhibiting a large systematic bias in their predictions might automatically perform better in the benchmark** due to their advantage in the AA1 task. Since we evaluate only aging-accelerating conditions, a positive systematic bias (where "positive" means that the predicted age acceleration tends to be statistically higher for healthy controls, whereas we expect it to be zero) should not be too large. **Such bias gives an unfair advantage to the model**. To account for this, we introduced a bracketed term in the BenchScore, which penalizes clocks with excessive systematic bias in their AA1 scores.
>
> Additionally, we provided a **full decomposition of our metric** in the form of Figures 3E,F and in Table 1. This allows for a detailed examination of each clock's performance. As the authors of the first aggregating metrics, we hope this work sparks active discussion and contributes to the development of more advanced metrics in the future.
>
> **Q3: It appears that plots C and D in Figure 3 may be incorrectly presented. Plot D should likely represent Med(|∆|) rather than Med(∆), as all points are above the diagonal. Please clarify if this is a mislabeling or if I have misunderstood the data shown.**
>
> No mislabeling here, but thank you for pointing to a potential issue with the readability of the figure. We will rewrite the caption in Fig 3C and D to be less confusing. The meaning of Chronological age prediction accuracy (Fig. 3C) is to measure the absolute error for each data sample (i.e. Med(|∆|)). In Chronological age prediction bias (Fig. 3D), we **measure the shift of the overall prediction** (i.e., Med(∆), as written in the paper) and it can be of negative or positive value. To better demonstrate the concept of “prediction bias” we sketched a limiting case in the Figure 3D **when all samples were predicted with a positive age acceleration**. This casts a strictly positive value of Med(∆), which is graphically represented as a red arrow on the figure.

---

### Author Response · Authors · 2024-11-20
**Rebuttal by Authors**

Dear Reviewers,

We sincerely thank you for your thoughtful and encouraging reviews. Your recognition of our work as *“likely to streamline method comparison and improve reliability in aging research”* (JKRR), *“well-defined and relevant for the domain”* (3vvy), *“a pleasure to read”* with metrics that *“foster progress in aging clock research”* (XgQd), and addressing key limitations in the field (PX5Y) inspires us greatly.

Several reviewers have expressed concerns regarding the relevance of our work to the ICLR community. In response, we emphasize that **biological age represents an interpretable latent variable** derived from primary biomarkers, with wide-ranging applications in computational and practical domains, including predicting mortality, assessing morbidity risk, and evaluating the efficacy of potential longevity interventions. Additionally, biological age **inherently lacks ground truth values**, that casts similarities with out-of-distribution (OOD) detection, unsupervised, self-supervised learning and other actively discussed topics at ICLR that heavily relies on data representations. We view our work as an invitation to the broader data science community to explore this challenging and impactful application of representation learning, **bridging the gap between clinical practice and machine learning**.

We deeply value your constructive suggestions and will work to incorporate them to further strengthen the manuscript. Please kindly refer to individual responses below for our point-by-point responses.
Thank you again for your time and insights. We are available for the follow-up discussion and further clarifications.

Best regards,

Authors of Submission 7894

---

### Author Response · Authors · 2024-11-26
**Submission update by Authors**

Dear Chairs and Reviewers,

We have updated the submission PDF to address the points raised in the Reviews. Below is the list of changes we have introduced:

1. In Benchmarking Methodology section 3.5, we have corrected the typo in the AA1 task description, replacing “**median** aging acceleration” with “**mean** aging acceleration”, as per our response to reviewer PX5Y.

2. In Benchmarking Methodology section 3.5, we have added the explicit mentions of the tasks: “Clearly, the first task (**AA2**) provides a more rigorous way to test aging clocks **compared to AA1**, because it helps to control potential covariate shifts, but the second task (**AA1**) deserves its place in the list, as it allows including more data into the panel to overcome data scarcity”, as per our response to reviewer XgQd.

3. In Benchmarking Methodology section 3.5, we replaced the last paragraph describing the 4th task (prediction bias task) with the following to include a deeper explanation of covariate shift, as per our response to reviewer PX5Y: “We introduced the fourth task, a prediction bias task, to evaluate the robustness of a given aging clock model to covariate shift between the original clock training dataset and the datasets from the proposed benchmark. Covariate shift, also referred to as batch effect in bioinformatics, denotes the shift between covariate distributions in two datasets. For instance, the distribution of methylation values for a given CpG site could be centered around 0.45 in one dataset and around 0.55 in the other one—a common scenario in DNAm and other omics data. Because each clock is trained on healthy controls, we expect age deviation of HC samples to be zero on average (*i.e.*, $E(\Delta_{HC})=0$). In practice, however, due to the presence of a covariate shift between the training and testing data, a clock might produce biased predictions, resulting in a systemic bias and adding or subtracting extra years for a healthy individual coming from an external dataset. The goal of the fourth task is to control for such systemic bias in clock predictions. Therefore, as a benchmarking metric for this task, we calculated median aging acceleration ($Med(\Delta)$) across HC samples from the entire dataset panel, which reflects the systematic shift in clock predictions caused by differences between datasets.”

4. In Benchmarking Methodology section 3.6, we added the following sentences before the last paragraph, addressing our discussion with reviewer PX5Y: “While designing our metric, we aimed for simplicity and interpretability. At the same time, we sought to include more data in the benchmark to address data scarcity caused by the underrepresentation of certain AACs.”

5. In the caption of Figure 3, we added the following, as per our response to reviewer PX5Y: “(C) illustrates that chronological age prediction accuracy is measured by median absolute error ($Med(|\Delta|)$) across all predictions. For a limiting case of prediction bias sketched in (D), all samples were predicted with positive age acceleration, leading to a strictly positive value of $Med(\Delta)$, graphically represented as a red arrow pointing to a cross.”

6. In Appendix section A.9, we clarified the rationale behind not performing any additional data procesing and inter-dataset normalization, as per our response to reviewer 3vvy, writing the following: “As there is no gold standard for DNAm pre-processing, each research group carries out their preferred pipeline that does not necessarily match the processing pipeline used for training the clock model, especially in case of applying earlier clocks (e.g., those by Hannum et al. (2013) or Horvath (2013)). Therefore, so as to retain this typical workflow and not to put any clock model into advantage by choosing the same processing that matches its own pipeline for every dataset, we did not perform any post-processing, inter-dataset normalization, or batch effect correction. In doing so, we also relied on two existing papers. First, compiling already pre-processed datasets without performing the same processing for all of them was done by Ying et al. (2023), another notable effort in the aging clock community. Second, we were also encouraged by a recent work by Varshavsky et al. (2023) who managed to create an accurate clock model by combining several blood datasets—without any additional normalization or correction procedure, using already pre-processed data from previous studies (some of which are included in our dataset as well), and thus demonstrating that the between-dataset normalization is not critical for this type of data.”

Best regards,

Authors of Submission 7894

---

### Author Response · Authors · 2024-11-28
**Post-discussion**

Dear reviewers,

As the discussion period is about to end, we kindly want to thank all reviewers for their time and useful comments. Also, please let us know if there are any other questions or comments, which we are ready to address promptly.

Our submission in a nutshell:
- Clear **criteria and methodology** of assessment tasks for comparing clock models
- Systematic curation of **66 public datasets**
- Comparison of 13 published biological aging clock models
- 19 health conditions across different ages and populations

Sincerely, Authors of submission 7894

---

### Meta-Review · Area_Chair_kHcT · 2024-12-22

**Metareview:**

The paper presents a framework for evaluating 13 published pre-trained linear regression-based biological clock models. Because no ground truth exists for biological age (a univariate latent representation), the paper presents a scoring mechanism to evaluate them on four related tasks.

**Strengths:**
 - The paper is very well written and easy to follow.
 - The compilation of the datasets is impressive and represents a thorough examination of how to compare epigenetic clock models.
 - The authors have participated heavily in the discussion period with lengthy responses to reviewer concerns.

**Weaknesses:**
 - The consistent and fundamental concern is whether this work is a good fit for the ICLR community. Put another way, it is not obvious whether the researchers who work on learning representations would benefit from this work. The models compared are all essentially the same linear regression model that have been pre-trained on different datasets. Several reviewers pointed out this flaw and remain unconvinced with the author rebuttal.
 - I want to acknowledge that the authors have made several arguments to defend the claim that the work is relevant to ICLR. They have stated that: a) biological age is inherently a learned representation lacking ground truth, b) they have cited several similar works published at comparable ML venues, and c) the key contribution of the work is not the compilation of datasets but instead "validating a clock's predictive ability to notice significant changes in patient's health, thus being a good indicator of biological age" (from author response to Reviewer 3vvy).
 - After close examination of the paper, reviewer comments, and author responses, I remain unconvinced by a) and b). For a), if learning biological age is a relevant representation-learning task, it is unclear why other methods from the ICLR community are not used. The authors list out-of-distribution (OOD) detection, unsupervised, and self-supervised learning as relevant methods, but none of these are used in the linear-regression methods that are benchmarked. For b), the listed similar works (in response to Reviewer XgQd) are indeed ML-venue benchmark and methodological papers on aging models using -omics data. However, the listed works focus much more on novel architectures or benchmarking more sophisticated models such as language models or generative models.
 - In response to c), the judgement about whether the work is clinically relevant is not the purview of ICLR. If this is indeed one of the main contributions of the work --- especially important in light of my thoughts on a) and b) --- then the work is better suited for a more relevant clinical venue.

The fit with ICLR's focus on methods, datasets, and benchmarks for learning representations weighed the most in my decision.

**Additional Comments On Reviewer Discussion:**

The paper had a very active discussion, and I commend both reviewers and authors for being engaged, respectful, and constructive in their responses.

Summary of the main points raised:
 1) **Reviewers pointed out small typos and clarifications (e.g., mean vs median)**: Authors acknowledged and addressed these comments
 2) **Reviewers noted that the main contribution of the paper is the compilation of datasets**: Authors maintain that the key contribution is the development of the evaluation scheme through the identification of relevant conditions and combination of the tasks. Note that the datasets compiled are all public. Two of the four benchmark tasks are previously developed (with citations). One is a "natural choice" given the first two tasks. The last one is also "well-known" (no citation).
3) **Reviewers asked about details about the evaluation of the clock models**: Authors confirmed that the clock models largely use the same architecture (i.e., a linear regression) and that the only difference across models is the training data used.

See above for my thought process in making my final decision.

---

### Decision · Program_Chairs · 2025-01-22

Reject